# Class-Prior Perturbation-Robust Regularization for Imbalanced Unreliable Partial Label Learning

Congyu Qiao [* 1 2]  Haohao Dong [* 1 2]  Xin Geng [1 2]  Ning Xu [1 2]

## Abstract

*Imbalanced Unreliable Partial Label Learning (I-UPLL)* is a challenging weakly supervised learning setting in which severe class imbalance and unreliable candidate labels jointly degrade model performance. By revisiting existing approaches for imbalanced learning, we observe that most of them fundamentally rely on estimating the class prior to guide balancing operations, such as re-sampling, pseudo-label generation, or logit adjustment. However, under I-UPLL, obtaining stable and accurate prior estimates at the early stage of training is often unrealistic due to the ambiguity and unreliability of partial labels, thereby leading the model to rapidly converge to a suboptimal solution. To address this issue, we propose CLAPOR, a novel *CLAss-PriOr perturbation-Robust regularization* framework that fundamentally avoids dependence on accurate prior estimation. Specifically, the proposed regularization trains the model under deliberately perturbed class priors, sampled from a Dirichlet distribution that deviates from the current estimated prior. This design encourages consistent performance under prior uncertainty and naturally preserves attention to minority classes. Extensive experiments on benchmark datasets demonstrate the effectiveness of CLAPOR across various settings of I-UPLL.

## 1. Introduction

Partial Label Learning (PLL) has emerged as a prominent weakly supervised learning paradigm, targeting multi-class classification tasks where each instance is annotated with a candidate label set instead of a unique ground-truth label (Hüllermeier & Beringer, 2006; Nguyen & Caruana, 2008). This paradigm naturally arises in crowdsourcing, web mining, and ambiguous annotation scenarios. With the advent of deep learning, PLL has advanced significantly, with methods exploring risk-consistent estimation (Feng et al., 2020; Wen et al., 2021), feature-level ground-truth label discrimination (Zhang et al., 2022; Wang et al., 2022b), and instance-label dependency modeling (Xu et al., 2021; Wu et al., 2024) to enhance performance.

To bridge the gap between theoretical PLL and real-world applications, recent works have relaxed its core assumptions, leading to more practical variants. On one hand, Unreliable Partial Label Learning (UPLL) proposed by (Lv et al., 2024) relaxes the label-level assumption that the ground-truth label is always contained in the candidate set, addressing annotation unreliability by refining reliable supervision signals or samples (Qiao et al., 2023; Wang et al., 2023; Xu et al., 2023a; Peng et al., 2025). On the other hand, Imbalanced Partial Label Learning (IPLL) proposed by (Wang et al., 2022a) relaxes the sample-level uniform prior assumption, tackling class imbalance via class-prior estimation to guide rebalancing, pseudo-label refinement, or logit adjustment (Wang et al., 2022a; Hong et al., 2023; Xu et al., 2024a; Jia et al., 2024; Huang et al., 2025).

Despite these advances, a critical gap remains: existing methods rarely consider the simultaneous relaxation of both label-level and sample-level assumptions, i.e., *Imbalanced Unreliable Partial Label Learning (I-UPLL)*. This scenario is not only more realistic but also introduces a fundamental dilemma faced by existing methods: annotation unreliability and ambiguity degrade class-prior estimation, while the inaccurately estimated class prior in turn distorts prior-dependent operations and hinders annotation refinement. As illustrated in Figure 1, this feedback loop causes the model to converge rapidly to a suboptimal solution in the early stage of training, particularly harming performance on minority classes.

To break this feedback loop, we propose CLAPOR, a novel *CLAss-PriOr perturbation-Robust regularization* framework based on bi-level optimization for I-UPLL. Unlike

*Equal contribution [1] School of Computer Science and Engineering, Southeast University, Nanjing 210096, China [2] Key Laboratory of New Generation Artificial Intelligence Technology and Its Interdisciplinary Applications (Southeast University), Ministry of Education, China. Email: {qiaocy,220255094,xgeng,xning}@seu.edu.cn. Correspondence to: Ning Xu <xning@seu.edu.cn>.

*Proceedings of the 43rd International Conference on Machine Learning*, Seoul, South Korea. PMLR 306, 2026. Copyright 2026 by the author(s).

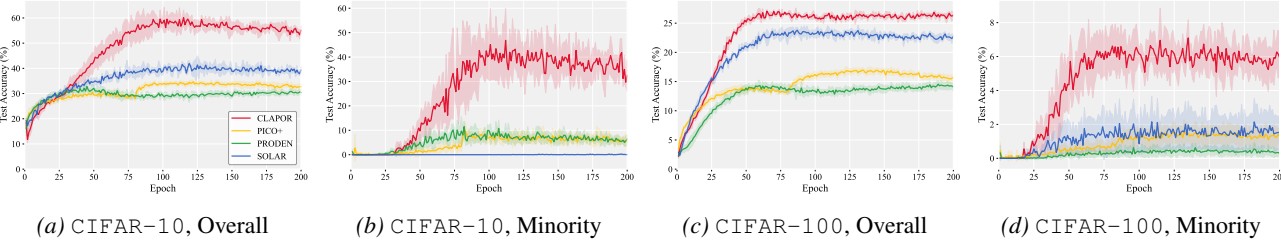

*(a)* CIFAR-10, Overall   *(b)* CIFAR-10, Minority   *(c)* CIFAR-100, Overall   *(d)* CIFAR-100, Minority

*Figure 1.* Training dynamics of overall and minority-class test accuracy for CLAPOR and baseline methods (PRODEN, PICO+, SOLAR) on CIFAR-10 ($\rho = 50, \gamma = 0.3, \iota = 0.3$) and CIFAR-100 ($\rho = 50, \gamma = 0.3, \iota = 0.1$). $\rho$ denotes class imbalance ratio, $\gamma$ controls label unreliability, and $\iota$ regulates label set ambiguity. CLAPOR maintains steady improvement while baselines collapse early, especially on minority classes.

existing methods that rely on accurate class-prior estimation, CLAPOR eliminates this dependency by enforcing the model to be robust to adversarial perturbations of class priors. This robustness is achieved via a novel regularization term that evaluates model performance under deliberately perturbed priors instantiated via Dirichlet sampling, encouraging consistent performance across diverse prior distributions and naturally preserving attention to minority classes. The bi-level optimization formulation of CLAPOR separates supervised learning and prior-robust regularization: the inner level minimizes a pseudo-label-based supervised loss to learn from available supervision, while the outer level minimizes the prior-robust regularization term. To implement this efficiently, we design an online training strategy with virtual parameters for gradient backpropagation, integrating iterative pseudo-label refinement via moving-average prior estimation, candidate label reconstruction, and two-stage gradient updates. This strategy ensures effectiveness in the extremely weakly supervised I-UPLL setting while maintaining computational efficiency. Overall, our contributions can be summarized as follows:

- We are the first to explicitly study Imbalanced Unreliable Partial Label Learning (I-UPLL), a realistic and challenging setting that relaxes both label-level and sample-level assumptions of traditional PLL. We propose a bi-level optimization-based regularized training framework that enables consistent model performance across diverse class priors, addressing the core challenge of unreliable prior estimation in I-UPLL.

- We design a powerful class-prior perturbation-robust regularization term, implemented via Dirichlet sampling to simulate adversarial prior perturbations. This term avoids reliance on accurate class-prior estimation, mitigates the negative impact of annotation noise on prior-dependent operations, and inherently preserves focus of the predictive model on minority classes.

The source code is available at https://github.com/palm-ml/clapor.

## 2. Related Work

Partial label learning (PLL) aims to learn a multi-class classifier from instances annotated with candidate label sets, where the correct label is concealed within each set (Hüllermeier & Beringer, 2006; Nguyen & Caruana, 2008). Existing PLL methods are generally categorized into two paradigms: average-based approaches (Hüllermeier & Beringer, 2006; Cour et al., 2011; Zhang & Yu, 2015) and identification-based approaches (Jin & Ghahramani, 2002; Liu & Dietterich, 2014; Zhang et al., 2016). With the seminal works of (Yao et al., 2020; Lv et al., 2020), PLL has formally entered the era of deep learning. Subsequently, (Feng et al., 2020; Wen et al., 2021) derive risk-consistent estimators for deep PLL, among which (Feng et al., 2020) further develops a classifier-consistent risk estimator. Beyond risk estimation, (Zhang et al., 2022; Wang et al., 2022b) aim to distinguish ground-truth labels through class activation value and prototype, while (Wu et al., 2022) refines supervision by leveraging the intrinsic consistency between instance features and candidate labels. In addition, (Xu et al., 2021; Qiao et al., 2022; Xu et al., 2023b; Wu et al., 2024; Xu et al., 2024c) advance PLL by explicitly modeling the dependency between candidate label sets and the underlying instances.

Recently, PLL has been extended toward more practical scenarios by progressively relaxing underlying assumptions.

By relaxing the label-level assumption that the correct label is always included in the candidate label set, (Lv et al., 2024) proposes unreliable partial label learning (UPLL) and addresses it by introducing a robust loss family. The learning paradigm is also referred to as Noisy Partial Label Learning (Wang et al., 2023; Peng et al., 2025). To mitigate label unreliability, (Qiao et al., 2023; Peng et al., 2025) reconstruct candidate label sets by including correct labels and reducing incorrect labels. (Wang et al., 2023) distinguishes reliable and unreliable samples and handles them in a semi-supervised contrastive framework. (Xu et al., 2023a) integrates supervision from model output and original annotation.

By relaxing the sample-level assumption that samples are

drawn from a uniform class prior, (Wang et al., 2022a) proposes imbalanced partial label learning (IPLL), which is also referred to as long-tailed partial label learning when the class distribution follows a long-tailed pattern (Hong et al., 2023; Jia et al., 2024). To handle class imbalance, these methods predominantly rely on estimating class priors to guide rebalancing operations. For instance, (Wang et al., 2022a) employs optimal transport constrained by estimated class priors, while (Xu et al., 2024a; Huang et al., 2025) regularize pseudo-label generation based on prior estimates. Moreover, (Hong et al., 2023) performs label disambiguation via dynamic logit adjustment informed by estimated class priors, and (Jia et al., 2024) decomposes the classifier into head and tail components and regularizes them through estimated class priors. Overall, we observe that class-prior estimation is a widely adopted strategy for handling class imbalance, not only in partial label learning but also across various weakly supervised learning settings (Menon et al., 2020; Wei & Gan, 2023; Zhou et al., 2024; Chen et al., 2025; Li et al., 2025).

In this paper, we consider simultaneously relaxing the assumptions at both the label level and the sample level, and study Imbalanced Unreliable Partial Label Learning (I-UPLL). The situation becomes fundamentally different, as annotation unreliability compromises the accuracy of class-prior estimation, which further propagates errors into the annotation refinement procedure. To address this issue, we propose a bi-level optimization framework equipped with a powerful regularization term that enforces robustness to adversarial perturbations of the class prior, naturally mitigating the negative effect of unstable and inaccurate class-prior estimation on the trained predictive model.

## 3. Proposed Method

### 3.1. Preliminaries

We first introduce some necessary notation. Let $\mathcal{X} = \mathbb{R}^q$ be the $q$-dimensional instance space and $\mathcal{Y} = \{1, 2, \ldots, c\}$ be the label space with $c$ class labels. Given the UPLL training set $\mathcal{D} = \{(\boldsymbol{x}_i, S_i) \,|\, 1 \leq i \leq n\}$ where $\boldsymbol{x}_i \in \mathcal{X}$ denotes the $q$-dimensional instance and $S_i \subset \mathcal{Y}$ denotes the candidate label set annotated to $\boldsymbol{x}_i$ with $S_i \notin \{\emptyset, \mathcal{Y}\}$. Unlike PLL, the correct label $y_i \in \mathcal{Y}$ of $\boldsymbol{x}_i$ may not be included in the candidate label set $S_i$. We introduce a logical matrix $\mathbf{S} = [\boldsymbol{s}_1, \boldsymbol{s}_2, \ldots, \boldsymbol{s}_n]^\top \in [0, 1]^{n \times c}$ to represent all candidate label sets, where $\boldsymbol{s}_i = [s_i^1, s_i^2, \ldots, s_i^c]$ maps to the candidate set $S_i$ of $\boldsymbol{x}_i$, with $s_i^j = 1$ indicating that class $j$ is annotated for $\boldsymbol{x}_i$ and 0 otherwise.

Further, we consider the imbalanced scenario by reformulating the UPLL training set as $\mathcal{D} = \cup_{j=1}^c \mathcal{D}^j$ with $\mathcal{D}^j = \{(\boldsymbol{x}_i, S_i) \,|\, 1 \leq i \leq n_j, y_i = j\}$ where $n^j = |\mathcal{D}^j|$ is the cardinality of instances of the $j$-th class in the train-

ing set. For the sake of clarity, we sort the classes by cardinality in decreasing order such that for the sequence $(n^1, n^2, \ldots, n^c)$, $n^1 \geq n^2 \geq \cdots \geq n^c$ and $n^1 \gg n^c$ in the imbalanced scenario. Given the class number threshold $\tau$, we define the minority classes as $\mathcal{Y}_{\min} = \{j \,|\, n^j \leq \tau\}$ independently. Similar to (Wang et al., 2022a; Hong et al., 2023), the task of UPLL with class-imbalance is to induce a multi-class classifier $f : \mathcal{X} \mapsto \mathcal{Y}$ from $\mathcal{D}$, taking into account both the overall classification accuracy $\text{Acc}(\mathcal{Y})$ and the minority class classification accuracy $\text{Acc}(\mathcal{Y}_{\min})$ on the unseen instances. Additionally, a parameterized score function $f(\cdot; \boldsymbol{\Theta}) : \mathcal{X} \mapsto \Delta^{c-1}$ is often employed as the prediction model where $\Delta^{c-1}$ denotes the $c$-dimensional simplex and its prediction on $\boldsymbol{x}$ is $\hat{y} = \arg\max_{j \in \mathcal{Y}} f^j(\boldsymbol{x}; \boldsymbol{\Theta})$.

In our proposed approach, we generate pseudo-labels $\mathbf{D} = [\boldsymbol{d}_1, \boldsymbol{d}_2, \ldots, \boldsymbol{d}_n]^\top$ as the supervision signal for the model $f(\cdot; \boldsymbol{\Theta})$, where $\boldsymbol{d}_i = [d_i^1, d_i^2, \ldots, d_i^c] \in \Delta^{c-1}$ denotes the generated pseudo-label for the instance $\boldsymbol{x}_i$, and the prediction model is often trained by the cross-entropy loss $\ell(\boldsymbol{d}, f(\boldsymbol{x}; \boldsymbol{\Theta})) = -\sum_{j=1}^c d^j \log f^j(\boldsymbol{x}; \boldsymbol{\Theta})$. The pseudo-label matrix $\mathbf{D}$ has been used as an important reference for estimating the prior class distribution of the training set $\boldsymbol{p}(y) = [p(y = 1), p(y = 2), \ldots, p(y = c)] \in \Delta^{c-1}$ in previous works (Wei et al., 2021; Lai et al., 2022; Xiao et al., 2022; Yang et al., 2023; Jia et al., 2024), or conversely, constrained by the estimated or assumed prior distribution $\hat{\boldsymbol{p}}(y)$ in previous works (Arazo et al., 2019; 2020; Wang et al., 2022a; Huang et al., 2025).

### 3.2. Overview

We propose CLAPOR, a prior-robust learning framework for UPLL that avoids relying on accurate class prior estimation. Unlike existing methods that explicitly estimate and exploit the class prior to guide pseudo-label generation or re-balancing, CLAPOR formulates learning as a bi-level optimization problem with a regularization term. The inner level minimizes a pseudo-label–based supervised loss, while the outer level introduces a regularization term that enforces robustness to arbitrary class priors. This regularization is constructed by evaluating model performance under deliberately perturbed prior distributions, implemented via Dirichlet sampling. As a result, the learned model is encouraged to perform consistently across diverse and potentially adversarial priors, naturally preserving attention to minority classes. CLAPOR is implemented through an online training strategy that alternates between supervised updates, prior-robust regularization, pseudo-label refinement, and candidate label reconstruction, making it effective and efficient in the extremely weakly supervised UPLL setting.

## 3.3. The CLAPOR Approach

In some state-of-the-art weakly supervised learning works, there is a tendency to use predictive model outputs $f(\boldsymbol{x}; \boldsymbol{\Theta})$ or pseudo-labels $\mathbf{D}$ to estimate the prior distribution $\boldsymbol{p}(y)$ with $\hat{\boldsymbol{p}}(y)$. Having a relatively accurate prior distribution estimation $\hat{\boldsymbol{p}}(y)$ is beneficial in multiple aspects: whether it is for re-sampling minority classes $\mathcal{Y}_{\text{few}}$ to balance the class distribution (Wang & Zhang, 2018), re-constraining the generation of pseudo-labels $\mathbf{D}$ using this prior distribution estimation $\hat{\boldsymbol{p}}(y)$ to produce more accurate pseudo-labels for model retraining (Wang et al., 2022a), or performing logit adjustment of the model (Hong et al., 2023).

However, things become very different in the UPLL scenario. Since the reliable supervision signals are extremely weak under the UPLL setting, it is very challenging to obtain a relatively accurate prior distribution estimation $\hat{\boldsymbol{p}}(y)$, leading the subsequent model training to converge rapidly to a suboptimal solution. Hence, to escape this suboptimal solution, we aim to construct a regularization term $\Psi(\boldsymbol{\Theta})$ for the supervised loss $\mathcal{L}(\boldsymbol{\Theta})$ that eliminates the need for accurate prior distribution estimation. When regularized by $\Psi(\boldsymbol{\Theta})$, the model becomes robust to the prior distribution from which the samples are derived. That is, the model can maintain acceptable accuracy on samples following any arbitrary prior distribution, thus naturally preserving its focus on the minority classes. Specifically, the overall objective is formulated by a bi-level optimization:

$$\min_{\boldsymbol{\Theta}} \Psi(\boldsymbol{\Theta}),$$
$$\text{s.t.} \quad \min_{\boldsymbol{\Theta}} \mathcal{L}(\boldsymbol{\Theta}, \mathcal{D}), \tag{1}$$

where the supervised loss $\mathcal{L}(\boldsymbol{\Theta})$ serves as a surrogate loss for the empirical risk $-\frac{1}{n}\sum_{i=1}^{n} y_i \log f^{y_i}(\boldsymbol{x}_i; \boldsymbol{\Theta})$, which is constructed by leveraging pseudo-labels $\mathbf{D}$ and defined as

$$\mathcal{L}(\boldsymbol{\Theta}, \mathcal{D}) = \frac{1}{n}\sum_{i=1}^{n} \ell(\boldsymbol{d}_i, f(\boldsymbol{x}_i; \boldsymbol{\Theta})). \tag{2}$$

The pseudo-labels $\mathbf{D}$ are initialized by

$$d_i^j = \begin{cases} \frac{1}{|S_i|} & \text{if } j \in S_i, \\ 0 & \text{otherwise,} \end{cases} \tag{3}$$

where $|\cdot|$ denotes the cardinality of a set. It is then updated by a dedicated algorithm $\mathcal{A}$ which is associated with the estimated prior $\hat{\boldsymbol{p}}(y)$ and the pseudo-label generation model $g(\cdot; \boldsymbol{\Omega})$, i.e.,

$$\mathbf{D} = \mathcal{A}(\mathcal{D}, \hat{\boldsymbol{p}}(y), \boldsymbol{\Omega}). \tag{4}$$

The core idea of our bi-level optimization Eq. (1) is to find the parameters that continue to minimize the regularization term $\Psi(\boldsymbol{\Theta})$ within the set of optimal parameters $\mathcal{S}(\boldsymbol{\Theta}^{\star}) = \{\boldsymbol{\Theta}^{\star}|\boldsymbol{\Theta}^{\star} = \arg\min_{\boldsymbol{\Theta}} \mathcal{L}(\boldsymbol{\Theta})\}$ for the supervised loss $\mathcal{L}(\boldsymbol{\Theta})$.

**Algorithm 1** The CLAPOR Approach

**Require:** The training dataset $\mathcal{D} = \{(\boldsymbol{x}_i, S_i)|1 \leq i \leq n\}$, total training epochs $T$, iterations per epoch $K$;
1: Initialize pseudo-labels $\mathbf{D}^{[0]}$ according to Eq. (3);
2: **for** $t = 1, 2, \ldots, T$ **do**
3:     **for** $k = 1, 2, \ldots, K$ **do**
4:         Create the virtual parameters $\boldsymbol{\Theta}_v^{[k-1]} = \boldsymbol{\Theta}^{[k-1]}$;
5:         Calculate $\mathcal{L}(\boldsymbol{\Theta}_v^{[k-1]}, \mathcal{I}_k)$ by Eq. (9) and update $\boldsymbol{\Theta}_v^{[k-1]}$ to $\boldsymbol{\Theta}_v^{[k]}$ by Eq. (10);
6:         Construct $\text{Dir}(\boldsymbol{\alpha}^{[t]})$ with its parameters initialized by Eq. (11) and sample mini-batch $\bar{\mathcal{I}}_k$ according to Eq. (12);
7:         Calculate $\Psi(\boldsymbol{\Theta}_v^{[k]}, \bar{\mathcal{I}}_k)$ by Eq. (13) and update $\boldsymbol{\Theta}^{[k-1]}$ to $\boldsymbol{\Theta}^{[k]}$ by Eq. (14);
8:     **end for**
9:     Generate pseudo-labels $\mathbf{D}^{[t]}$ according to Eq. (4);
10:    Estimate class prior $\hat{\boldsymbol{p}}^{[t]}(y)$ of training samples according to Eq. (7);
11:    Reconstruct candidate labels and update $\mathcal{D}^{[t-1]}$ to $\mathcal{D}^{[t]}$ by Eq. (8);
12: **end for**
**Ensure:** The predictive model $f(\cdot; \boldsymbol{\Theta})$.

Next, we explore how to construct such a $\Psi(\boldsymbol{\Theta})$ that renders the model insensitive to the prior distribution. Since the prior $\boldsymbol{p}(y) \in \Delta^{c-1}$, it can be rewritten as a categorical distribution parameterized by $\boldsymbol{z} = [z^1, z^2, \ldots, z^c] \in \Delta^{c-1}$, i.e., $\text{Cat}(\boldsymbol{z})$, and $p(y = j) = z^j$. We assume that the parameters $\boldsymbol{z}$ of the prior $\boldsymbol{p}(y)$ are drawn from an unknown distribution parameterized by $\boldsymbol{\alpha}$, i.e., $\boldsymbol{z} \sim \boldsymbol{q}(\boldsymbol{\alpha})$, which, intuitively, motivates us to design a regularization term $\Psi(\boldsymbol{\Theta})$ that enhances the model's robustness to prior uncertainty:

$$\Psi(\boldsymbol{\Theta}) = \mathbb{E}_{\substack{\boldsymbol{z} \sim \boldsymbol{q}(\boldsymbol{\alpha}), \\ y \sim \text{Cat}(\boldsymbol{z}), \\ \boldsymbol{x} \sim p(\boldsymbol{x}|y)}} \left[ \mathbf{1}\left\{ y \neq \arg\max_{j \in \mathcal{Y}} f^j(\boldsymbol{x}; \boldsymbol{\Theta}) \right\} \right], \tag{5}$$

where $\mathbf{1}\{\cdot\}$ denotes an indicator function that takes value 1 if the condition holds, and 0 otherwise. By optimizing $\Psi(\boldsymbol{\Theta})$ in Eq. (5), the model's average prediction error rate on samples is expected to be minimized across all possible prior distributions.

Similar to $\mathcal{L}(\boldsymbol{\Theta})$, since the correct label $y$ is unknown, we also employ pseudo-labels $\mathbf{D}$ as supervision signal and reformulate Eq. (5) as:

$$\Psi(\boldsymbol{\Theta}) = \mathbb{E}_{\substack{\boldsymbol{z} \sim \boldsymbol{q}(\boldsymbol{\alpha}), y \sim \text{Cat}(\boldsymbol{z}), \\ \boldsymbol{x} \sim p(\boldsymbol{x}|y)}} \left[ \ell(\boldsymbol{d}, f(\boldsymbol{x}; \boldsymbol{\Theta})) \right], \tag{6}$$

where the sampled $y = \arg\max_{j \in \mathcal{Y}} d_i^j$.

In practice, we design an online training strategy to solve the objective Eq. (1) for deriving our final predictive model

*Table 1.* The overall test accuracy of each comparing approach (mean ± std) under various I-UPLL settings of `CIFAR-10`.

| $\rho$ | 50 | | | | 100 | | | | 200 | | | |
|---|---|---|---|---|---|---|---|---|---|---|---|---|
| $(\gamma, \iota)$ | (0.3,0.3) | (0.3,0.5) | (0.4,0.2) | (0.5,0.3) | (0.3,0.3) | (0.3,0.5) | (0.4,0.2) | (0.5,0.3) | (0.3,0.3) | (0.3,0.5) | (0.4,0.2) | (0.5,0.3) |
| PRODEN | 33.51±0.61 | 27.03±0.98 | 34.35±0.67 | 25.89±0.49 | 31.44±0.47 | 25.32±0.82 | 31.79±1.18 | 25.51±1.07 | 29.64±0.44 | 25.15±0.53 | 30.68±1.06 | 24.32±0.82 |
| VALEN | 30.48±0.44 | 24.37±0.63 | 32.92±0.94 | 25.06±0.73 | 29.69±0.70 | 23.67±0.64 | 30.68±0.61 | 24.57±0.44 | 27.74±0.55 | 22.32±1.02 | 28.67±0.50 | 24.35±0.78 |
| CC | 33.44±1.25 | 26.00±0.54 | 33.70±1.14 | 25.46±0.68 | 31.09±1.20 | 25.19±0.86 | 31.20±0.97 | 24.38±1.23 | 29.65±0.67 | 23.45±0.57 | 30.34±0.81 | 23.86±0.31 |
| LW | 35.41±1.45 | 19.23±2.69 | 32.58±1.30 | 21.28±1.86 | 32.54±1.32 | 18.50±2.29 | 30.39±1.18 | 22.20±2.15 | 29.79±0.59 | 17.43±0.37 | 29.56±0.34 | 23.69±0.47 |
| PICO | 32.77±0.97 | 30.28±1.68 | 32.42±0.72 | 24.03±0.59 | 29.81±0.46 | 26.64±0.92 | 29.73±1.09 | 23.14±0.63 | 27.99±0.71 | 25.95±0.96 | 28.15±0.85 | 22.72±0.78 |
| SOLAR | 43.72±1.60 | 29.86±2.15 | 37.85±0.99 | 24.34±1.01 | 37.76±0.63 | 29.61±1.33 | 35.10±0.39 | 22.94±0.38 | 34.71±1.02 | 27.75±0.32 | 31.87±0.57 | 22.43±0.48 |
| ROBUSTL | 32.75±0.29 | 25.90±0.79 | 35.09±0.56 | 26.22±0.91 | 31.18±0.73 | 24.16±0.56 | 31.76±0.79 | 24.88±0.64 | 29.89±0.91 | 23.33±0.35 | 30.42±0.73 | 24.35±0.87 |
| PICO+ | 35.44±0.68 | 27.56±0.90 | 36.43±0.70 | 27.40±0.61 | 32.18±0.81 | 26.25±0.61 | 33.05±0.84 | 25.59±0.59 | 30.47±0.70 | 24.76±1.20 | 30.99±0.38 | 24.17±0.58 |
| ALIM | 30.06±0.39 | 25.01±0.49 | 31.92±0.83 | 24.86±0.98 | 29.01±0.45 | 23.96±0.74 | 30.42±0.47 | 23.90±0.62 | 27.21±0.69 | 23.10±0.60 | 28.57±0.49 | 23.53±0.70 |
| PSCR | 30.30±0.61 | 24.24±0.56 | 31.76±0.91 | 24.95±0.51 | 28.98±0.64 | 24.09±0.49 | 29.76±0.82 | 23.84±0.64 | 28.08±0.63 | 23.18±0.92 | 28.06±0.70 | 23.16±0.81 |
| CLAPOR | **61.68±2.80** | **42.32±2.90** | **59.29±0.28** | **34.86±2.09** | **53.35±1.82** | **35.85±1.01** | **50.03±2.09** | **31.17±0.83** | **46.35±0.82** | **34.17±0.99** | **43.62±0.16** | **29.64±1.27** |

$f(\cdot; \boldsymbol{\Theta})$. To begin with, given a total of $T$ training epochs, we generate pseudo-labels $\mathbf{D}^{[t]}$ for the training set $\mathcal{D}^{[t]}$ at the $t$-th epoch via Eq. (4). Here, the pseudo-label generation algorithm $\mathcal{A}$ is instantiated as the excellent strategy proposed in (Wang et al., 2022a). We employ the predictive model directly as the pseudo-label generation model, i.e., $f(\cdot; \boldsymbol{\Theta}) = g(\cdot; \boldsymbol{\Omega})$. The estimated class prior $\hat{p}(y)$ is initialized with a uniform distribution $\hat{\boldsymbol{p}}^{[0]}(y) = \left[\frac{1}{c}, \frac{1}{c}, \ldots, \frac{1}{c}\right]$ and updated using a moving-average scheme, i.e.,

$$\hat{\boldsymbol{p}}^{[t]}(y) = \mu \hat{\boldsymbol{p}}^{[t-1]}(y) + (1-\mu)\frac{1}{n}\sum_{i=1}^{n} f(\boldsymbol{x}_i; \boldsymbol{\Theta}), \qquad (7)$$

where $\mu \in [0,1]$ denotes a weight hyperparameter that controls the update rate of the prior.

To adapt to UPLL, candidate label sets in the training set are also required to be updated by a reconstruction strategy $\mathcal{B}$ at the end of every epoch:

$$\mathcal{D}^{[t]} = \mathcal{B}(\mathcal{D}^{[t-1]}, \mathbf{D}^{[t]}), \qquad (8)$$

where we instantiate $\mathcal{B}$ as the state-of-the-art reconstruction strategy proposed by (Peng et al., 2025).

Then, in every iteration $k$ of the $t$-th training epoch, we shuffle and transform the training set $\mathcal{D}$ into $K$ mini-batches $\{\mathcal{I}_1, \mathcal{I}_2, \ldots, \mathcal{I}_K\}$, each containing $m$ training samples $\mathcal{I}_k = \{(\boldsymbol{x}_i, \boldsymbol{d}_i) | 1 \leq i \leq m\}$. Following (Finn et al., 2017; Xu et al., 2024b), we create a kind of virtual parameters $\boldsymbol{\Theta}_v^{[k-1]} = \boldsymbol{\Theta}^{[k-1]}$ for differentiation and the batch-style supervised loss $\mathcal{L}(\boldsymbol{\Theta}_v^{[k-1]}, \mathcal{I}_k)$ is calculated by

$$\mathcal{L}(\boldsymbol{\Theta}_v^{[k-1]}, \mathcal{I}_k) = \frac{1}{m}\sum_{i=1}^{m} \ell(\boldsymbol{d}_i, f(\boldsymbol{x}_i; \boldsymbol{\Theta}_v^{[k-1]})). \qquad (9)$$

We update the virtual parameters from $\boldsymbol{\Theta}_v^{[k-1]}$ to $\boldsymbol{\Theta}_v^{[k]}$ through:

$$\boldsymbol{\Theta}_v^{[k]} = \boldsymbol{\Theta}_v^{[k-1]} - \eta_1 \frac{\partial \mathcal{L}(\boldsymbol{\Theta}_v^{[k-1]}, \mathcal{I}_k)}{\partial \boldsymbol{\Theta}_v^{[k-1]}}, \qquad (10)$$

where $\eta_1$ is the step size.

After the optimization through Eq. (10), $\boldsymbol{\Theta}_v^{[k]}$ will depend on $\boldsymbol{\Theta}_v^{[k-1]}$, which we can explicitly express as $\boldsymbol{\Theta}_v^{[k]}(\boldsymbol{\Theta}_v^{[k-1]})$. In this way, we can obtain the final gradient of $\Psi(\boldsymbol{\Theta}_v^{[k]}(\boldsymbol{\Theta}_v^{[k-1]}))$ with respect to $\boldsymbol{\Theta}_v^{[k-1]}$, and apply the gradient to update $\boldsymbol{\Theta}^{[k-1]}$ to $\boldsymbol{\Theta}^{[k]}$.

To implement this optimization process, we propose a practical instantiation strategy that replaces the intractable unknown prior $\boldsymbol{q}(\boldsymbol{\alpha})$ with a constructed Dirichlet prior $\text{Dir}(\boldsymbol{\alpha})$ with $\boldsymbol{\alpha} = [\alpha_1, \alpha_2, \ldots, \alpha_c]$ and $\alpha > 0$, which ensures that the prior parameters $\boldsymbol{z}$ sampled from $\text{Dir}(\boldsymbol{\alpha})$ deviate from the estimated prior $\hat{\boldsymbol{p}}^{[t]}(y)$ to simulate our desired robustness in Eq. (6). Intuitively, we can set

$$\boldsymbol{\alpha}^{[t]} = \lambda \cdot \text{Reverse}(\hat{\boldsymbol{p}}^{[t]}(y)) + \beta, \qquad (11)$$

where the operation $\text{Reverse}(\cdot)$ reverses the order of the vector sorted by the magnitude of its elements, and constants $\lambda > 0$ and $\beta > 0$ control the stability and smoothness of Dirichlet sampling. According to the properties of the Dirichlet distribution, we have $\mathbb{E}[z^j] = \frac{\alpha^j}{\sum_{j=1}^{c} \alpha^j}$, thereby leading the sampled $\boldsymbol{z}$ to adversarially deviate from $\hat{\boldsymbol{p}}^{[t]}(y)$. Meanwhile, different parameters $\boldsymbol{z}$ introduce a prior perturbation when sampling classes from it.

After specifying $\boldsymbol{q}(\boldsymbol{\alpha})$, we can first sample prior parameters $\boldsymbol{z}^{[t]}$ from $\boldsymbol{q}(\boldsymbol{\alpha})$, and then sample a training batch $\bar{\mathcal{I}}_k$ for $\Psi(\boldsymbol{\Theta})$ with batch size equal to $m$ and each $(\boldsymbol{x}, \boldsymbol{d}) \in \bar{\mathcal{I}}_k$ sampled according to:

$$\begin{aligned} y &\sim \text{Cat}(\boldsymbol{z}^{[t]}), \\ (\boldsymbol{x}, \boldsymbol{d}) &\sim \{(\boldsymbol{x}, \boldsymbol{d}) | \arg\max_{j \in \mathcal{Y}} d^j = y\}. \end{aligned} \qquad (12)$$

The batch-style regularization term $\Psi(\boldsymbol{\Theta}_v^{[k]}, \bar{\mathcal{I}}_k)$ is calculated by

$$\Psi(\boldsymbol{\Theta}_v^{[k]}, \bar{\mathcal{I}}_k) = \frac{1}{m}\sum_{i=1}^{m} \ell(\boldsymbol{d}_i, f(\boldsymbol{x}_i; \boldsymbol{\Theta}_v^{[k]})). \qquad (13)$$

Finally, we apply the gradients of the regularization term $\Psi(\boldsymbol{\Theta}_v^{[k]})$ and the supervision term $\mathcal{L}(\boldsymbol{\Theta}_v^{[k-1]}, \mathcal{I}_k)$ with respect to $\boldsymbol{\Theta}_v^{[k-1]}$ to our predictive model $f(\cdot; \boldsymbol{\Theta})$ at the $k$-th

*Table 2.* The overall test accuracy of each comparing approach (mean ± std) under various I-UPLL settings of `CIFAR-100`.

| $\rho$ | 10 | | | | 20 | | | | 50 | | | |
|---|---|---|---|---|---|---|---|---|---|---|---|---|
| $(\gamma,\iota)$ | (0.3,0.05) | (0.3,0.1) | (0.5,0.05) | (0.5,0.1) | (0.3,0.05) | (0.3,0.1) | (0.5,0.05) | (0.5,0.1) | (0.3,0.05) | (0.3,0.1) | (0.5,0.05) | (0.5,0.1) |
| PRODEN | 27.19±0.33 | 21.54±0.38 | 18.48±0.39 | 14.05±0.39 | 23.76±0.22 | 18.38±0.55 | 16.11±0.46 | 11.85±0.56 | 19.49±0.12 | 14.99±0.51 | 13.77±0.34 | 10.12±0.23 |
| VALEN | 22.79±0.56 | 17.11±0.38 | 16.78±0.43 | 12.34±0.72 | 19.93±0.44 | 15.40±0.33 | 14.72±0.48 | 10.94±0.25 | 17.11±0.55 | 13.12±0.30 | 12.85±0.33 | 9.67±0.35 |
| CC | 27.75±0.89 | 22.41±0.50 | 18.16±0.25 | 13.69±0.39 | 24.42±0.66 | 18.94±0.14 | 15.77±0.40 | 11.73±0.61 | 20.12±0.46 | 15.87±0.70 | 13.35±0.39 | 10.01±0.27 |
| LW | 24.76±0.70 | 16.82±0.33 | 13.42±0.58 | 9.59±0.41 | 21.04±0.36 | 14.50±0.51 | 12.49±0.37 | 8.96±0.35 | 17.71±0.38 | 12.58±0.53 | 11.53±0.32 | 8.71±0.36 |
| PICO | 30.37±0.51 | 27.23±0.44 | 22.14±0.32 | 18.33±0.35 | 25.34±0.28 | 23.23±1.02 | 18.85±0.45 | 15.92±0.38 | 20.96±0.60 | 19.33±0.44 | 15.35±0.21 | 13.46±0.23 |
| SOLAR | 41.24±0.57 | 36.37±0.84 | 28.67±0.74 | 21.54±1.15 | 35.67±0.28 | 30.62±0.70 | 23.28±0.53 | 18.81±1.13 | 29.52±0.69 | 24.49±0.47 | 18.89±0.27 | 15.30±0.57 |
| ROBUSTL | 21.93±0.75 | 18.37±1.29 | 14.42±0.60 | 11.90±0.56 | 18.25±0.75 | 15.49±0.79 | 12.70±0.38 | 10.56±0.52 | 15.06±0.40 | 12.92±0.54 | 11.25±0.43 | 9.76±0.29 |
| PICO+ | 31.49±0.48 | 23.18±0.51 | 21.32±0.50 | 15.08±0.61 | 27.40±0.48 | 20.80±0.46 | 18.50±0.80 | 13.31±0.61 | 22.69±0.49 | 17.27±0.47 | 14.85±0.48 | 10.94±0.38 |
| ALIM | 17.93±0.28 | 13.46±0.41 | 13.32±0.40 | 9.21±0.56 | 15.46±0.74 | 11.88±0.17 | 11.65±0.29 | 8.00±0.23 | 13.50±0.42 | 10.13±0.45 | 9.98±0.08 | 7.29±0.29 |
| PSCR | 18.37±0.41 | 13.39±0.45 | 13.57±0.22 | 9.27±0.39 | 15.81±0.40 | 11.60±0.25 | 11.52±0.57 | 8.34±0.38 | 13.33±0.16 | 10.30±0.40 | 10.20±0.31 | 6.95±0.36 |
| **CLAPOR** | **43.14±0.40** | **38.94±0.58** | **30.72±0.83** | **25.14±0.39** | **37.25±0.37** | **33.45±0.62** | **26.20±1.20** | **20.54±0.11** | **30.54±0.10** | **27.93±0.07** | **21.28±0.30** | **16.90±0.80** |

*Table 3.* The minority-class test accuracy of each comparing approach (mean ± std) under various I-UPLL settings of `CIFAR-10`.

| $\rho$ | 50 | | | | 100 | | | | 200 | | | |
|---|---|---|---|---|---|---|---|---|---|---|---|---|
| $(\gamma,\iota)$ | (0.3,0.3) | (0.3,0.5) | (0.4,0.2) | (0.5,0.3) | (0.3,0.3) | (0.3,0.5) | (0.4,0.2) | (0.5,0.3) | (0.3,0.3) | (0.3,0.5) | (0.4,0.2) | (0.5,0.3) |
| PRODEN | 16.05±1.88 | 13.46±8.08 | 20.18±1.85 | 19.40±5.50 | 12.05±2.13 | 14.66±9.63 | 15.42±0.53 | 14.12±3.20 | 8.95±1.27 | 8.11±3.35 | 14.28±3.05 | 14.27±2.46 |
| VALEN | 12.09±1.36 | 6.35±3.85 | 16.61±0.93 | 13.41±1.22 | 8.85±2.15 | 4.45±2.08 | 14.52±1.56 | 14.16±0.97 | 7.68±1.22 | 3.73±1.48 | 12.78±1.31 | 13.89±1.55 |
| CC | 22.64±2.33 | 17.98±2.94 | 24.54±1.24 | 19.84±1.59 | 17.10±2.68 | 13.99±4.34 | 20.90±2.45 | 17.12±1.32 | 13.74±0.96 | 13.27±2.25 | 16.89±2.25 | 20.16±3.17 |
| LW | 0.49±0.69 | 4.44±5.09 | 5.13±4.16 | 11.72±4.03 | 0.01±0.01 | 1.61±1.77 | 1.11±0.97 | 11.63±7.20 | 0.00±0.00 | 4.23±5.61 | 0.96±0.60 | 7.10±2.56 |
| PICO | 27.12±2.35 | **25.93±3.68** | 27.16±3.43 | 25.89±3.24 | 21.44±2.31 | 22.06±1.25 | 21.38±2.40 | 21.12±0.89 | 18.54±3.26 | 17.82±5.33 | 18.94±0.84 | 19.76±2.24 |
| SOLAR | 4.98±1.73 | 0.42±0.23 | 7.10±1.68 | 0.74±0.29 | 1.62±1.15 | 0.67±0.72 | 5.74±0.61 | 0.62±0.16 | 2.82±2.11 | 2.24±2.45 | 2.31±1.52 | 0.47±0.33 |
| ROBUSTL | 21.12±2.21 | 19.05±1.44 | 22.14±1.93 | 20.33±3.43 | 19.39±1.79 | 19.16±1.45 | 19.41±2.30 | 19.41±3.09 | 17.50±1.61 | 17.96±2.20 | 17.73±1.15 | 17.44±1.29 |
| PICO+ | 12.61±2.56 | 12.96±3.66 | 18.64±2.94 | 24.96±5.23 | 9.11±1.89 | 22.14±7.70 | 15.57±2.95 | 14.30±3.93 | 8.66±4.17 | 18.17±9.00 | 10.52±2.24 | 16.36±4.86 |
| ALIM | 19.41±2.66 | 25.18±5.83 | 20.59±1.46 | 22.55±3.44 | 18.72±2.30 | **24.31±6.93** | 19.58±3.45 | 25.87±4.20 | 19.83±6.05 | 21.10±5.73 | 19.40±2.75 | 21.49±2.90 |
| PSCR | 16.12±5.33 | 21.11±6.01 | 15.57±3.42 | 15.89±4.16 | 14.17±3.00 | 18.93±8.17 | 12.53±2.81 | 19.47±5.37 | 7.00±3.07 | **22.64±5.78** | 14.01±2.78 | 18.19±4.88 |
| **CLAPOR** | **52.40±9.86** | 21.41±2.16 | **48.83±2.79** | **32.94±1.01** | **29.88±7.28** | 23.78±5.02 | **28.08±3.91** | **28.50±13.31** | **20.60±2.19** | 9.40±2.19 | **20.32±4.82** | **30.53±5.37** |

iteration:

$$
\begin{aligned}
\Theta^{[k]} =\ & \Theta^{[k-1]} - \eta_1 \frac{\partial \mathcal{L}(\Theta_v^{[k-1]}, \mathcal{I}_k)}{\partial \Theta_v^{[k-1]}} \\
& - \eta_2 \frac{\partial \Psi(\Theta_v^{[k]}, \bar{\mathcal{I}}_k)}{\partial \Theta_v^{[k-1]}}.
\end{aligned}
\tag{14}
$$

where $\eta_2$ is the step size.

The whole algorithmic description of CLAPOR is presented in Algorithm 1. CLAPOR focuses on regularization for samples with deviations from the current class distribution, thus imposing low requirements on the accuracy of prior estimation. This inherently preserves focus on minority classes and avoids suboptimal convergence while exhibiting excellent compatibility with the UPLL setting with scarce reliable supervision. Practically, CLAPOR adopts an online strategy to implement this kind of regularization in the bi-level optimization framework, ensuring effectiveness and efficiency.

## 4. Experiment

### 4.1. Datasets

We evaluate our CLAPOR approach on standard multi-class classification benchmark datasets, including `CIFAR-10` and `CIFAR-100` (Krizhevsky et al., 2009), which are transformed to support flexible control of data imbalance, label unreliability, and ambiguity.

As for data imbalance, we follow (Cao et al., 2019; Wei et al., 2021) and randomly remove the training images class by class to satisfy a given imbalanced ratio $\rho = \frac{n^1}{n^c}$, where $n^1 = 5000$ in `CIFAR-10` and $n^1 = 500$ in `CIFAR-100`. Different imbalanced ratios are employed for evaluation. We control $\rho \in \{50, 100, 200\}$ for `CIFAR-10`. Due to fewer instances per class than `CIFAR-10`, we control $\rho \in \{10, 20, 50\}$ for `CIFAR-100`. For the sake of convenience, class indices are sorted in descending order according to the sample size of each class.

As for label unreliability and ambiguity, we adopt a similar generation method introduced in (Qiao et al., 2023; Wang et al., 2023; Peng et al., 2025) to further manually corrupt the imbalanced dataset into the UPLL case. Each candidate label set $S_i$ of the instance $\boldsymbol{x}_i$ is synthesized from a potentially unreliable label $\tilde{y}_i$ and a candidate set $\tilde{S}_i$ that does not contain the ground-truth label, i.e., $\{\tilde{y}_i\} \cup \tilde{S}_i$. The potentially unreliable label $\tilde{y}_i$ is controlled by an unreliable-label rate $\gamma$ and sampled from a categorical distribution $\mathrm{Cat}(\tilde{\boldsymbol{z}}_i)$ with $\tilde{\boldsymbol{z}}_i = [\tilde{z}_i^1, \tilde{z}_i^2, \ldots, \tilde{z}_i^c] \in \Delta^{c-1}$ where $\tilde{z}_i^{y_i} = 1 - \gamma$ and $\tilde{z}_i^{j \neq y_i} = \frac{\gamma}{c-1}$. The candidate set $\tilde{S}_i$ that does not contain the ground-truth label is controlled by a partial rate $\iota$ and sampled from a multivariate Bernoulli distribution $\mathrm{Ber}(\tilde{\boldsymbol{l}}_i)$ with $\tilde{\boldsymbol{l}}_i = [\tilde{l}_i^1, \tilde{l}_i^2, \ldots, \tilde{l}_i^c] \in [0,1]^c$ where $\tilde{l}_i^{y_i} = 0$ and $\tilde{l}_i^{j \neq y_i} = \iota$. For `CIFAR-10`, the controlled constants $(\gamma, \iota)$ vary in $\{(0.3, 0.3), (0.3, 0.5), (0.4, 0.2), (0.5, 0.3)\}$. For `CIFAR-100`, due to the larger label space, they are controlled in $\{(0.3, 0.05), (0.3, 0.1), (0.5, 0.05), (0.5, 0.1)\}$.

On each version of the datasets, we run three trials with dif-

*Table 4.* The minority-class test accuracy of each comparing approach (mean ± std) under various I-UPLL settings of `CIFAR-100`.

| $\rho$ | 10 | | | | 20 | | | | 50 | | | |
|---|---|---|---|---|---|---|---|---|---|---|---|---|
| $(\gamma, \iota)$ | (0.3,0.05) | (0.3,0.1) | (0.5,0.05) | (0.5,0.1) | (0.3,0.05) | (0.3,0.1) | (0.5,0.05) | (0.5,0.1) | (0.3,0.05) | (0.3,0.1) | (0.5,0.05) | (0.5,0.1) |
| PRODEN | 14.75±1.12 | 7.28±1.03 | 7.83±0.82 | 3.61±0.49 | 9.23±0.67 | 2.78±0.71 | 4.46±0.67 | 1.80±0.57 | 3.55±0.36 | 0.90±0.34 | 2.69±0.49 | 1.01±0.43 |
| VALEN | 8.68±0.32 | 4.28±0.86 | 5.66±0.78 | 1.76±0.39 | 5.35±0.42 | 2.04±0.58 | 3.01±0.57 | 1.08±0.22 | 2.80±0.81 | 0.76±0.42 | 1.82±0.33 | 0.53±0.27 |
| CC | 15.87±0.69 | 9.68±0.51 | 8.27±0.57 | 5.01±1.12 | 9.63±0.96 | 4.43±0.51 | 4.75±0.58 | 2.76±0.59 | 4.02±0.91 | 1.48±0.43 | 2.49±0.60 | 1.01±0.45 |
| LW | 5.41±1.68 | 0.38±0.22 | 1.78±1.44 | 0.57±0.47 | 0.96±0.61 | 0.06±0.09 | 1.02±0.37 | 0.49±0.25 | 0.18±0.20 | 0.00±0.00 | 0.19±0.18 | 0.06±0.04 |
| PICO | 19.72±1.30 | 14.44±1.24 | 11.09±0.78 | 7.86±0.97 | 12.12±0.44 | 8.84±1.11 | 6.96±0.71 | 4.08±0.88 | 5.65±0.64 | 4.11±0.51 | 3.25±0.45 | 2.95±0.97 |
| SOLAR | 25.27±0.61 | 17.78±2.48 | 12.10±0.07 | 7.40±1.28 | 15.31±0.12 | 10.45±1.15 | 6.10±0.40 | 4.66±0.82 | 5.99±0.61 | 2.74±1.64 | 1.51±0.58 | 0.82±0.73 |
| ROBUSTL | 2.04±1.12 | 3.49±0.75 | 0.73±0.62 | 1.63±1.31 | 0.00±0.00 | 0.82±0.74 | 0.57±0.98 | 0.68±0.73 | 0.00±0.00 | 0.24±0.53 | 0.00±0.00 | 0.32±0.45 |
| PICO+ | 18.99±0.91 | 9.91±1.62 | 12.44±0.56 | 6.38±1.13 | 11.81±0.85 | 5.63±1.17 | 6.81±0.97 | 4.20±0.98 | 5.31±0.48 | 1.92±0.56 | 2.91±0.71 | 1.83±0.35 |
| ALIM | 8.04±0.83 | 5.33±0.62 | 5.21±0.37 | 3.64±0.49 | 5.08±0.56 | 3.68±0.26 | 4.14±0.52 | 2.82±0.27 | 3.05±0.23 | 2.56±0.37 | 2.40±0.27 | 1.95±0.09 |
| PSCR | 7.57±0.61 | 5.32±0.37 | 5.44±0.76 | 3.47±0.63 | 5.27±0.99 | 3.03±0.33 | 3.19±0.44 | 2.76±0.22 | 2.74±0.38 | 1.98±0.25 | 2.09±0.24 | 1.71±0.40 |
| CLAPOR | **30.21±0.26** | **25.13±1.18** | **18.20±1.20** | **13.64±0.78** | **20.36±1.22** | **15.21±1.04** | **10.33±1.42** | **6.82±0.88** | **9.31±0.56** | **7.85±1.16** | **4.68±1.30** | **3.90±1.26** |

*Table 5.* Summary of the Wilcoxon signed-rank test for CLAPOR against other comparing approaches at 0.05 significance level. The *p*-values are shown in the brackets.

| CLAPOR against | | PRODEN | VALEN | CC | LW | PICO | SOLAR | ROBUSTL | PICO+ | ALIM | PSCR |
|---|---|---|---|---|---|---|---|---|---|---|---|
| CIFAR-10 | Overall | **win**[0.0005] | **win**[0.0005] | **win**[0.0005] | **win**[0.0005] | **win**[0.0005] | **win**[0.0005] | **win**[0.0005] | **win**[0.0005] | **win**[0.0005] | **win**[0.0005] |
| | Minority | **win**[0.0005] | **win**[0.0005] | **win**[0.0024] | **win**[0.0005] | **win**[0.0342] | **win**[0.0005] | **win**[0.0049] | **win**[0.0034] | tie[0.0771] | **win**[0.0068] |
| CIFAR-100 | Overall | **win**[0.0005] | **win**[0.0005] | **win**[0.0005] | **win**[0.0005] | **win**[0.0005] | **win**[0.0005] | **win**[0.0005] | **win**[0.0005] | **win**[0.0005] | **win**[0.0005] |
| | Minority | **win**[0.0005] | **win**[0.0005] | **win**[0.0005] | **win**[0.0005] | **win**[0.0005] | **win**[0.0005] | **win**[0.0005] | **win**[0.0005] | **win**[0.0005] | **win**[0.0005] |

ferent random seeds, record the best comprehensive performance within the total epochs, and report the mean accuracy and standard deviation of all compared algorithms.

## 4.2. Baselines

We compare the performance of CLAPOR with 10 state-of-the-art baseline methods in closely related fields including PLL, IPLL, and UPLL:

- PRODEN (Lv et al., 2020), which progressively identifies correct labels by re-weighting the model outputs on candidate labels.

- VALEN (Xu et al., 2021), which builds a variational inference model to infer correct labels to deal with instance-dependent candidate labels.

- CC (Feng et al., 2020), which derives a classifier-consistent risk to deal with PLL.

- LW (Wen et al., 2021), which weighs candidate and non-candidate labels through a leverage parameter in the proposed loss to deal with class-dependent candidate labels.

- PICO (Wang et al., 2022b), which proposes an additional contrastive loss term to enhance the disambiguation ability of the predictive model in PLL.

- SOLAR (Wang et al., 2022a), which proposes an optimal-transport-based framework that refines disambiguated labels to align with the marginal class prior distribution.

- ROBUSTL (Lv et al., 2024), which leverages the proposed average partial-label loss family with rigorous theoretical guarantees of robustness in the UPLL setting. In our experiments, we select the Mean Squared Error (ROBUSTL) as the representative loss function.

- PICO+ (Wang et al., 2023), which extends PICO to UPLL by implementing clean sample detection and processing the remaining noisy samples within a semi-supervised contrastive learning paradigm.

- ALIM (Xu et al., 2023a), which yields pseudo-labels via balancing the initial candidate set and model outputs, and then performing normalization for supervision. In our experiments, we select the normalization based on temperature as the representative.

- PSCR (Peng et al., 2025), which proposes a metric termed ECK to distinguish between clean and noisy samples, and reconstructs candidate label sets to reduce their size while ensuring the inclusion of ground-truth labels.

We adopt the same prediction model for fair comparisons. Specifically, ResNet-18 (He et al., 2016) is trained on `CIFAR-10` and ConvNet (Liu et al., 2022) is trained on `CIFAR-100`, where Stochastic Gradient Descent (SGD) is used with the learning rate and momentum set to 0.01 and 0.9, respectively. We set the maximum epoch to 1000 and train until model performance converges. The batch size is 256, and the learning rate decays through the cosine learning rate schedule. These configurations are applied for CLAPOR and all baselines for fair comparisons, while other hyperparameters of all baselines are selected as suggested in the respective literature. We also apply common data augmentations: Random Horizontal Flipping, Random Cropping, Cutout (Devries & Taylor, 2017), and AutoAugment (Cubuk et al., 2019). All comparison experiments are conducted using PyTorch on NVIDIA GeForce RTX 4090 D GPUs.

*Table 6.* Test accuracy (mean ± std) of CLAPOR and its variant CLAPOR-N under various I-UPLL settings of benchmark datasets.

| | | CIFAR-10 | | | | | | CIFAR-100 | | | |
| | | Overall | | Minority | | | | Overall | | Minority | |
| $\rho$ | $(\gamma,\iota)$ | CLAPOR | CLAPOR-N | CLAPOR | CLAPOR-N | $\rho$ | $(\gamma,\iota)$ | CLAPOR | CLAPOR-N | CLAPOR | CLAPOR-N |
|---|---|---|---|---|---|---|---|---|---|---|---|
| 50 | (0.3,0.3) | **61.68±2.80** | 48.15±2.63 | **52.40±9.86** | 17.69±7.06 | 10 | (0.3,0.05) | **43.14±0.40** | 41.08±0.13 | **30.21±0.26** | 25.50±0.27 |
| | (0.3,0.5) | **42.32±2.90** | 35.87±0.73 | **21.41±2.16** | 5.24±2.22 | | (0.3,0.1) | **38.94±0.58** | 35.13±0.83 | **25.13±1.18** | 18.67±1.88 |
| | (0.4,0.2) | **59.29±0.28** | 45.31±1.41 | **48.83±2.79** | 13.14±3.15 | | (0.5,0.05) | **30.72±0.83** | 28.81±0.32 | **18.20±1.20** | 13.13±1.49 |
| | (0.5,0.3) | **34.86±2.09** | 26.59±0.43 | **32.94±1.01** | 1.00±0.22 | | (0.5,0.1) | **25.14±0.39** | 19.94±0.11 | **13.64±0.78** | 7.91±1.34 |
| 100 | (0.3,0.3) | **53.35±1.82** | 42.07±1.14 | **29.88±7.28** | 6.32±0.33 | 20 | (0.3,0.05) | **37.25±0.37** | 35.64±0.33 | **20.36±1.22** | 16.49±1.30 |
| | (0.3,0.5) | **35.85±1.01** | 31.49±0.48 | **23.78±5.02** | 3.70±1.77 | | (0.3,0.1) | **33.45±0.62** | 30.71±0.89 | **15.21±1.04** | 10.68±0.52 |
| | (0.4,0.2) | **50.03±2.09** | 37.60±1.00 | **28.08±3.91** | 5.57±0.73 | | (0.5,0.05) | **26.20±1.20** | 23.66±0.40 | **10.33±1.42** | 6.07±0.76 |
| | (0.5,0.3) | **31.17±0.83** | 24.63±0.70 | **28.50±13.31** | 1.04±0.52 | | (0.5,0.1) | **20.54±0.11** | 16.26±0.16 | **6.82±0.88** | 4.04±1.25 |
| 200 | (0.3,0.3) | **46.35±0.82** | 38.38±2.07 | **20.60±2.19** | 0.86±0.22 | 50 | (0.3,0.05) | **30.54±0.10** | 29.27±0.49 | **9.31±0.56** | 6.87±1.69 |
| | (0.3,0.5) | **34.17±0.99** | 29.97±1.45 | **9.40±2.19** | 1.53±0.51 | | (0.3,0.1) | **27.93±0.07** | 24.75±0.11 | **7.85±1.16** | 4.81±1.23 |
| | (0.4,0.2) | **43.62±0.16** | 35.39±0.74 | **20.32±4.82** | 1.30±0.21 | | (0.5,0.05) | **21.28±0.30** | 18.74±0.71 | **4.68±1.30** | 2.73±0.84 |
| | (0.5,0.3) | **29.64±1.27** | 24.06±0.55 | **30.53±5.37** | 0.71±0.07 | | (0.5,0.1) | **16.90±0.80** | 14.27±0.34 | **3.90±1.26** | 1.76±0.46 |

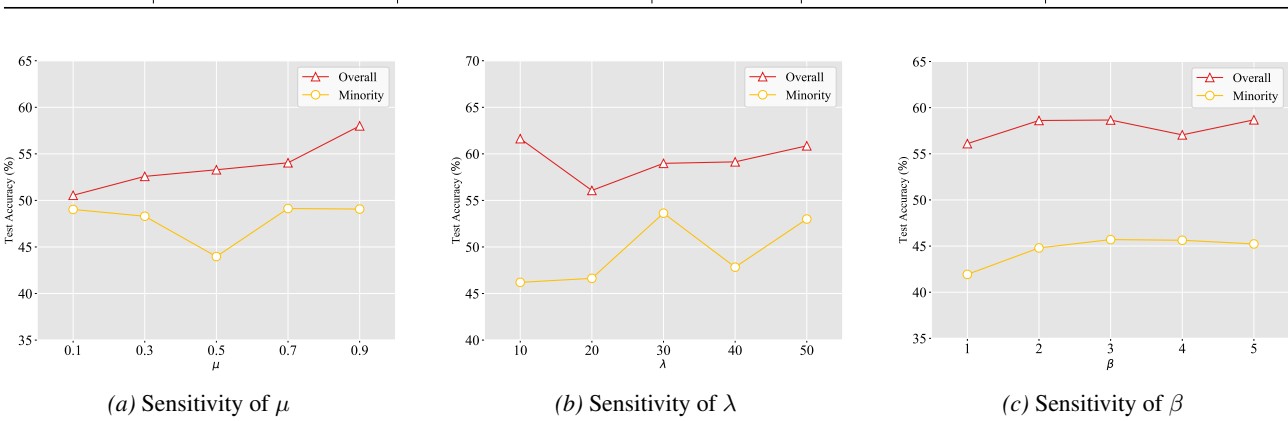

*(a) Sensitivity of $\mu$*      *(b) Sensitivity of $\lambda$*      *(c) Sensitivity of $\beta$*

*Figure 2.* The parameter sensitivity analysis for CLAPOR on CIFAR-10 when $\rho = 50$, $\gamma = 0.3$, and $\iota = 0.3$.

### 4.3. Experimental Results

Tables 1 and 2 report the overall classification accuracy $\text{Acc}(\mathcal{Y})$ and standard deviation of each compared algorithm under various I-UPLL settings of benchmark datasets, while Tables 3 and 4 report the minority-class classification accuracy $\text{Acc}(\mathcal{Y}_{\min})$ and standard deviation. Note that since class indices are sorted in descending order according to the sample size, we set $\mathcal{Y}_{\min} = \{8, 9, 10\}$ for CIFAR-10 and $\mathcal{Y}_{\min} = \{68, 69, \dots, 100\}$ for CIFAR-100. The best results are highlighted in bold. Furthermore, the Wilcoxon signed-rank test is employed to show whether CLAPOR significantly outperforms other compared approaches. Table 5 reports the $p$-values for the corresponding tests and the statistical test results at the 0.05 significance level. From these tables, we can observe:

- CLAPOR consistently achieves the best overall performance across all settings on both benchmark datasets.

- CLAPOR obtains the best minority-class performance under most settings on CIFAR-10 and all settings on CIFAR-100.

- CLAPOR significantly outperforms all competing baselines in terms of both overall and minority-class accuracy, except for a tie with ALIM on CIFAR-10.

### 4.4. Further Analysis

To further validate the helpfulness of the class-prior perturbation-robust regularization in CLAPOR, we conduct an ablation study using a vanilla variant of our method, denoted CLAPOR-N, where this regularization term is ablated and the predictive model is optimized solely via the pseudo-label-based supervised loss. As shown in Table 6, CLAPOR outperforms CLAPOR-N across all experimental settings of I-UPLL, which confirms that the class-prior perturbation-robust regularization plays a critical role in boosting the performance of the predictive model for I-UPLL tasks.

Figures 2a, 2b, and 2c illustrate the performance of CLAPOR under different values of $\mu$, $\lambda$, and $\beta$ on CIFAR-10 when $\rho = 50$, $\gamma = 0.3$, and $\iota = 0.3$. We vary $\mu$ in $[0.1, 0.9]$, $\lambda$ in $[10, 50]$, and $\beta$ in $[1, 5]$. As shown, CLAPOR exhibits stable performance across a wide range of hyper-parameter settings. This robustness is highly desirable, as it indicates that CLAPOR can achieve consistent and reliable classification

performance without sensitive hyper-parameter tuning.

## 5. Conclusion

In this paper, we formally study Imbalanced Unreliable Partial Label Learning (I-UPLL), a realistic yet largely unexplored weakly supervised learning setting where class imbalance and annotation unreliability coexist. We identify that the heavy reliance on class-prior estimation in existing methods leads to a harmful feedback loop under unreliable supervision, resulting in premature suboptimal convergence and degraded minority-class performance. To address this issue, we propose CLAPOR, a class-prior perturbation–robust regularization framework that fundamentally avoids dependence on accurate prior estimation by enforcing robustness to deliberately perturbed priors. Extensive experimental results demonstrate that CLAPOR consistently improves both overall and minority-class performance across diverse I-UPLL settings of benchmark datasets.

## Acknowledgement

This research was supported by the National Science Foundation of China (624B2040, 62576093, 62125602, U24A20324, and 92464301), the Jiangsu Science Foundation (BG2024036), the Fundamental Research Funds for the Central Universities (2242025K30024), the New Cornerstone Science Foundation through the XPLORER PRIZE, and the Big Data Computing Center of Southeast University.

## Impact Statement

This paper presents work whose goal is to advance the field of machine learning. There are many potential societal consequences of our work, none of which we feel must be specifically highlighted here.

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

*Table 7.* The majority-class test accuracy of each comparing approach (mean ± std) under various I-UPLL settings of `CIFAR-10`.

| $\rho$ | 50 | | | | 100 | | | | 200 | | | |
|---|---|---|---|---|---|---|---|---|---|---|---|---|
| $(\gamma, \iota)$ | (0.3,0.3) | (0.3,0.5) | (0.4,0.2) | (0.5,0.3) | (0.3,0.3) | (0.3,0.5) | (0.4,0.2) | (0.5,0.3) | (0.3,0.3) | (0.3,0.5) | (0.4,0.2) | (0.5,0.3) |
| PRODEN | 78.49±2.12 | 71.78±1.42 | 78.34±0.94 | 69.73±1.76 | 78.13±1.31 | 72.06±0.90 | 78.04±0.66 | 72.12±1.66 | 78.01±1.63 | 71.42±0.87 | 78.37±0.75 | 69.41±1.18 |
| VALEN | 78.07±0.98 | 70.82±0.65 | 78.53±1.40 | 71.82±2.15 | 78.06±1.58 | 71.43±1.84 | 79.19±1.51 | 70.24±1.86 | 76.23±2.03 | 67.71±2.50 | 79.51±0.28 | 69.32±1.29 |
| CC | 77.83±1.54 | 69.84±3.02 | 77.55±1.35 | 66.71±2.75 | 78.57±1.19 | 67.97±1.48 | 77.64±1.27 | 65.87±2.82 | 77.07±1.28 | 68.68±3.40 | 79.16±0.67 | 66.86±2.52 |
| LW | 85.61±0.54 | 64.09±8.98 | 83.35±0.96 | 62.95±1.80 | 85.44±0.78 | 61.65±7.64 | 84.34±0.71 | 66.37±2.05 | 83.94±0.65 | 58.09±1.22 | 83.80±0.89 | 71.98±0.90 |
| PICO | 73.90±1.53 | 62.24±2.10 | 74.73±1.38 | 61.25±2.80 | 74.68±1.92 | 60.79±3.04 | 74.49±1.46 | 63.07±1.83 | 74.57±0.79 | 61.88±2.37 | 74.77±1.31 | 62.23±2.73 |
| SOLAR | **90.99±0.57** | **86.11±1.35** | **89.12±0.10** | 76.37±1.67 | **91.53±0.61** | **84.66±2.22** | **89.52±0.29** | 76.44±1.29 | **90.31±0.46** | **88.37±0.99** | **88.29±0.38** | 74.78±1.59 |
| ROBUSTL | 77.62±1.04 | 67.05±3.62 | 78.14±0.73 | 69.39±1.82 | 78.05±0.72 | 68.77±2.15 | 77.73±1.90 | 69.80±1.62 | 77.24±0.85 | 67.16±1.44 | 78.31±1.48 | 68.55±1.22 |
| PICO+ | 76.44±1.05 | 71.56±1.57 | 76.69±1.04 | 70.37±0.92 | 77.12±0.82 | 71.04±3.10 | 76.33±0.53 | 69.39±1.41 | 76.92±1.11 | 70.31±0.99 | 77.14±1.44 | 69.01±1.00 |
| ALIM | 75.89±1.44 | 69.08±0.88 | 76.91±1.91 | 67.34±2.78 | 77.07±1.28 | 67.50±2.27 | 76.55±1.22 | 64.24±1.96 | 75.59±1.29 | 67.37±1.43 | 77.49±1.17 | 67.71±2.29 |
| PSCR | 75.78±1.28 | 67.58±2.34 | 76.40±0.98 | 68.92±2.18 | 76.17±0.89 | 68.86±2.49 | 76.85±1.97 | 69.49±1.51 | 77.10±1.61 | 65.78±3.96 | 76.10±1.11 | 67.99±2.13 |
| CLAPOR | 85.27±0.50 | 72.37±2.31 | 87.57±0.83 | **76.51±0.75** | 86.67±0.54 | 73.10±2.17 | 85.96±1.07 | **76.62±1.28** | 85.20±1.42 | 73.84±2.10 | 83.83±0.26 | **77.50±1.07** |

*Table 8.* The medium-class test accuracy of each comparing approach (mean ± std) under various I-UPLL settings of `CIFAR-10`.

| $\rho$ | 50 | | | | 100 | | | | 200 | | | |
|---|---|---|---|---|---|---|---|---|---|---|---|---|
| $(\gamma, \iota)$ | (0.3,0.3) | (0.3,0.5) | (0.4,0.2) | (0.5,0.3) | (0.3,0.3) | (0.3,0.5) | (0.4,0.2) | (0.5,0.3) | (0.3,0.3) | (0.3,0.5) | (0.4,0.2) | (0.5,0.3) |
| PRODEN | 34.14±0.97 | 24.91±1.50 | 34.85±1.24 | 25.48±1.00 | 29.39±1.30 | 20.92±1.25 | 30.93±1.37 | 23.55±2.54 | 26.00±1.41 | 21.70±3.12 | 29.33±1.02 | 21.48±1.38 |
| VALEN | 26.58±0.78 | 20.54±4.57 | 30.31±1.64 | 22.72±2.97 | 24.59±2.46 | 16.14±4.14 | 27.03±2.53 | 20.25±1.59 | 21.24±1.79 | 14.96±2.91 | 24.34±1.63 | 21.89±1.15 |
| CC | 32.08±1.85 | 25.31±1.23 | 32.97±1.38 | 25.39±1.65 | 29.80±2.36 | 24.23±3.19 | 31.39±2.25 | 24.62±1.64 | 28.45±2.94 | 19.47±1.50 | 29.57±1.49 | 22.50±2.04 |
| LW | 32.18±4.28 | 5.75±6.47 | 29.25±2.18 | 23.96±2.75 | 24.39±2.81 | 2.16±2.07 | 23.89±1.77 | 21.23±1.25 | 18.93±2.25 | 3.67±4.65 | 20.83±0.73 | 20.73±4.97 |
| PICO | 36.72±2.36 | 30.27±1.92 | 33.46±1.12 | 26.84±1.13 | 33.36±0.60 | 26.48±1.47 | 30.83±1.96 | 24.72±1.10 | 30.80±1.58 | 26.15±1.78 | 28.66±1.89 | 26.57±2.10 |
| SOLAR | 44.42±4.67 | 14.39±4.46 | 33.45±1.67 | 8.33±0.54 | 33.38±2.81 | 14.78±3.79 | 25.69±0.20 | 8.06±0.80 | 24.90±3.07 | 6.15±1.46 | 22.02±1.04 | 5.14±0.23 |
| ROBUSTL | 32.66±2.17 | 24.48±1.45 | 36.02±1.27 | 24.38±1.65 | 28.68±2.38 | 22.49±0.98 | 31.18±1.33 | 23.75±1.75 | 28.01±1.46 | 20.50±0.85 | 27.88±1.25 | 22.45±1.83 |
| PICO+ | 33.57±2.20 | 22.34±2.94 | 35.51±1.16 | 25.55±2.88 | 28.12±1.27 | 20.44±3.42 | 30.17±1.08 | 23.08±1.38 | 26.35±1.49 | 18.31±4.64 | 28.10±1.12 | 21.38±2.64 |
| ALIM | 27.93±1.59 | 25.41±1.33 | 31.05±0.69 | 26.07±1.51 | 27.33±2.09 | 22.96±0.70 | 28.57±1.39 | 23.97±2.41 | 24.09±1.67 | 23.90±3.83 | 25.94±1.52 | 24.54±2.27 |
| PSCR | 29.97±3.45 | 24.39±2.25 | 32.04±1.74 | 25.13±2.41 | 25.36±1.85 | 25.31±0.70 | 27.53±2.37 | 24.46±2.08 | 25.13±0.56 | 23.48±2.57 | 26.14±2.05 | 23.31±2.83 |
| CLAPOR | **64.24±1.95** | **48.08±5.88** | **63.42±0.45** | **36.41±5.94** | **60.89±2.17** | **37.15±6.97** | **54.68±1.30** | **32.29±3.48** | **51.01±2.37** | **39.72±0.98** | **49.41±1.29** | **27.05±0.92** |

# A. Appendix

## A.1. Additional Experimental Results

Tables 7–10 report test accuracy on majority and medium classes with different class-frequency groups. For `CIFAR-10`, the set of majority classes is set as $\{1, 2, 3\}$ and the set of medium classes is set as $\{4, 5, 6, 7\}$. For `CIFAR-100`, the set of majority classes is set as $\{1, 2, \ldots, 33\}$ and the set of medium classes is set as $\{34, 35, \ldots, 67\}$. From the tables, we can observe that CLAPOR achieves competitive or superior performance on all I-UPLL settings of benchmark datasets.

**Reverse() Construction Ablation.** To verify the rationale behind our proposed *Reverse*($\cdot$) operation for the Dirichlet concentration parameter, we compare it with three alternative prior construction strategies. As illustrated in Table 11, CLAPOR-Uniform sets $\alpha = \mathbf{1}$ and assigns equal weights regardless of the estimated prior, CLAPOR-Random samples from $\alpha_j = \lambda \cdot u_j + \beta$ with $u_j \sim \text{Uniform}(0, 1)$, CLAPOR-Direct directly uses the estimated prior as $\alpha = \lambda \cdot \hat{p} + \beta$, and CLAPOR uses our strategy $\alpha = \lambda \cdot Reverse(\hat{p}) + \beta$. CLAPOR achieves the highest minority-class accuracy (62.26%), which is 8.13 percentage points higher than CLAPOR-Direct. This gap is substantially larger than the difference in overall accuracy (1.04 percentage points), suggesting that the *Reverse*($\cdot$) operation better protects minority-class attention. Notably, CLAPOR significantly outperforms CLAPOR-Uniform, confirming that inverting the estimated prior is more effective than a uniform prior when the estimate is inaccurate. Even under unreliable labels where the estimated prior is potentially biased, *Reverse*($\cdot$) still produces informative perturbations that prevent the model from overfitting to possibly wrong prior estimates while encouraging attention to minority classes.

**Extended Sensitivity of $\beta$.** To verify the robustness of CLAPOR to the Dirichlet concentration parameter $\beta$, we extend the sensitivity analysis beyond the range $[1, 5]$ reported in the main paper. Table 12 reports the performance across $\beta \in \{0.25, 0.5, 1, 2, 3, 4, 5, 10, 20, 30, 40, 50\}$ on `CIFAR-10` ($\rho = 50, \gamma = 0.3, \iota = 0.3$). While CLAPOR exhibits reasonable tolerance to $\beta$, the performance is not entirely flat: setting $\beta$ too small (e.g., 0.25 or 0.5) degrades both overall and minority-class accuracy, while a very large $\beta$ (e.g., 40 or 50) also harms performance. Among the displayed settings, when $\beta \approx 20$, CLAPOR achieves the best trade-off between minority-class recognition (58.10%) and overall accuracy (62.28%). We therefore recommend $\beta = 20$ as a reasonable default, with modest tuning around this value for specific datasets.

**Performance with Reliable Labels ($\gamma = 0$).** To isolate the contribution of the prior-robust regularization from the challenges specific to unreliable partial labels, we evaluate CLAPOR under the setting where candidate labels are fully reliable

*Table 9.* The majority-class test accuracy of each comparing approach (mean ± std) under various I-UPLL settings of `CIFAR-100`.

| $\rho$ | 10 | | | | 20 | | | | 50 | | | |
|---|---|---|---|---|---|---|---|---|---|---|---|---|
| $(\gamma, \iota)$ | (0.3,0.05) | (0.3,0.1) | (0.5,0.05) | (0.5,0.1) | (0.3,0.05) | (0.3,0.1) | (0.5,0.05) | (0.5,0.1) | (0.3,0.05) | (0.3,0.1) | (0.5,0.05) | (0.5,0.1) |
| PRODEN | 45.75±0.86 | 40.54±0.80 | 34.20±0.37 | 28.50±0.54 | 44.22±0.31 | 39.50±0.98 | 33.80±0.54 | 27.66±1.46 | 41.85±0.53 | 37.02±0.84 | 33.35±0.80 | 27.06±0.47 |
| VALEN | 41.94±0.84 | 35.56±0.72 | 33.05±0.72 | 27.06±0.80 | 41.76±0.59 | 35.08±0.55 | 32.45±0.62 | 27.69±0.58 | 40.10±0.78 | 34.07±0.59 | 31.91±0.74 | 26.71±0.49 |
| CC | 44.13±0.81 | 39.05±0.25 | 33.72±0.19 | 27.11±0.81 | 43.60±0.62 | 38.52±0.86 | 33.38±0.46 | 27.48±0.81 | 42.26±1.03 | 37.17±1.33 | 32.27±0.63 | 27.17±0.96 |
| LW | 47.32±0.45 | 38.64±1.14 | 29.76±1.49 | 24.24±1.29 | 46.13±0.90 | 36.89±0.87 | 30.00±1.18 | 24.52±1.13 | 44.30±1.02 | 36.09±0.63 | 30.91±0.49 | 25.30±0.63 |
| PICO | 46.32±0.53 | 43.61±0.84 | 37.84±1.10 | 33.55±0.84 | 44.22±0.66 | 41.56±1.49 | 35.78±0.77 | 33.58±0.79 | 41.73±0.73 | 40.49±0.93 | 33.43±0.98 | 32.09±1.39 |
| SOLAR | 59.22±0.57 | 56.99±0.70 | 47.70±0.39 | 40.72±1.75 | 58.66±1.03 | 54.74±1.29 | 46.04±0.19 | 40.08±1.69 | 57.23±0.23 | 53.64±0.42 | 43.26±0.07 | 38.66±0.84 |
| ROBUSTL | 45.45±0.49 | 39.41±1.34 | 33.49±0.89 | 28.29±0.63 | 43.70±0.58 | 37.95±0.55 | 32.86±0.89 | 28.17±1.11 | 41.17±0.97 | 36.85±0.71 | 31.79±1.32 | 28.04±0.46 |
| PICO+ | 45.48±0.98 | 38.59±0.67 | 33.80±0.89 | 28.52±1.09 | 44.70±0.66 | 39.08±1.10 | 33.09±1.22 | 27.49±0.46 | 43.09±0.82 | 38.01±0.97 | 32.43±0.34 | 26.21±0.57 |
| ALIM | 34.13±0.54 | 26.53±0.61 | 26.30±0.63 | 19.61±1.33 | 32.78±1.31 | 27.46±0.87 | 25.50±0.80 | 18.62±0.99 | 31.96±0.88 | 26.25±1.26 | 25.05±0.95 | 19.48±0.95 |
| PSCR | 33.42±1.14 | 26.92±0.74 | 27.42±1.05 | 19.75±1.01 | 32.43±0.91 | 26.78±1.00 | 25.59±0.41 | 20.30±0.58 | 31.85±0.75 | 26.46±0.92 | 25.67±0.65 | 18.58±1.24 |
| CLAPOR | 57.94±0.45 | 56.55±1.18 | 45.07±0.76 | 39.21±1.30 | 58.22±0.40 | 54.55±0.26 | 44.65±1.18 | 38.22±0.79 | 57.19±1.97 | **55.24±0.78** | 42.16±1.28 | 36.92±2.68 |

*Table 10.* The medium-class test accuracy of each comparing approach (mean ± std) under various I-UPLL settings of `CIFAR-100`.

| $\rho$ | 10 | | | | 20 | | | | 50 | | | |
|---|---|---|---|---|---|---|---|---|---|---|---|---|
| $(\gamma, \iota)$ | (0.3,0.05) | (0.3,0.1) | (0.5,0.05) | (0.5,0.1) | (0.3,0.05) | (0.3,0.1) | (0.5,0.05) | (0.5,0.1) | (0.3,0.05) | (0.3,0.1) | (0.5,0.05) | (0.5,0.1) |
| PRODEN | 28.91±0.46 | 21.79±0.67 | 17.96±0.95 | 13.47±0.82 | 24.18±0.94 | 16.68±0.87 | 14.67±1.06 | 9.08±1.44 | 17.54±0.60 | 11.05±0.82 | 9.79±0.40 | 5.38±0.14 |
| VALEN | 22.81±0.39 | 14.12±1.00 | 14.96±0.57 | 10.21±1.42 | 17.86±0.82 | 11.63±0.78 | 12.01±0.47 | 6.61±0.57 | 13.88±0.95 | 7.87±0.83 | 8.29±0.86 | 4.12±0.90 |
| CC | 28.96±1.07 | 22.74±0.87 | 17.82±1.37 | 12.93±0.75 | 24.35±0.29 | 18.99±0.73 | 14.41±0.77 | 9.03±0.84 | 19.05±1.33 | 12.52±1.50 | 10.42±0.61 | 6.13±0.92 |
| LW | 24.32±1.26 | 13.54±1.32 | 11.07±0.82 | 5.28±0.91 | 18.52±1.49 | 7.58±1.18 | 9.19±1.26 | 3.15±1.13 | 10.04±0.63 | 2.53±1.31 | 5.39±1.06 | 2.25±1.04 |
| PICO | 32.45±1.06 | 28.33±0.80 | 22.22±0.98 | 18.72±0.99 | 26.46±0.82 | 24.76±0.67 | 18.30±1.06 | 15.35±1.03 | 19.98±0.61 | 18.09±0.29 | 13.30±1.06 | 10.67±1.04 |
| SOLAR | 43.55±1.00 | 39.14±0.57 | 29.49±1.91 | 20.86±1.02 | 37.39±0.64 | 30.79±1.32 | 22.31±0.75 | 15.69±1.91 | 29.08±2.18 | 20.10±1.83 | 14.13±0.80 | 8.86±1.38 |
| ROBUSTL | 21.38±1.27 | 15.18±2.58 | 11.69±1.81 | 8.06±1.71 | 13.85±2.35 | 9.51±2.69 | 6.45±1.43 | 5.02±1.10 | 6.36±1.75 | 4.13±1.46 | 2.92±1.53 | 2.61±1.05 |
| PICO+ | 32.61±0.37 | 23.62±0.83 | 21.36±0.98 | 15.26±0.69 | 28.30±0.72 | 20.07±1.20 | 18.49±0.86 | 12.03±0.84 | 22.07±0.75 | 14.07±1.17 | 12.63±0.97 | 7.68±1.51 |
| ALIM | 17.51±0.61 | 12.97±1.10 | 13.18±0.77 | 8.67±0.83 | 13.91±0.64 | 9.53±1.09 | 10.05±0.95 | 6.87±0.76 | 10.48±0.68 | 6.96±0.74 | 7.05±0.44 | 4.39±0.60 |
| PSCR | 18.30±1.21 | 12.46±0.91 | 13.36±0.99 | 8.39±0.46 | 14.55±0.52 | 9.39±0.96 | 9.98±0.74 | 6.37±0.58 | 10.25±0.67 | 6.27±0.32 | 6.55±0.73 | 4.31±0.56 |
| CLAPOR | **45.83±0.26** | **41.86±0.77** | **32.02±0.96** | **26.04±1.21** | **39.60±0.18** | **36.21±0.80** | **27.49±1.37** | **20.72±1.07** | **32.43±1.08** | **27.69±0.94** | **21.07±0.70** | **13.79±0.97** |

($\gamma = 0$, imbalanced PLL only). Table 13 compares CLAPOR against four baselines on `CIFAR-10` ($\rho = 100, \iota = 0.3$). CLAPOR still significantly outperforms all competing methods with an overall accuracy of 82.21% and a minority-class accuracy of 75.67%, demonstrating that the proposed prior-robust regularization benefits imbalanced label learning in general, not only under unreliable supervision.

**Architecture Comparison.** To address the concern regarding the use of different architectures for `CIFAR-10` (ResNet-18) and `CIFAR-100` (ConvNet), we ablate representative backbone choices for both datasets. As shown in Table 14, CLAPOR is consistently evaluated with both ResNet-18 and ConvNet on both datasets. On `CIFAR-10`, ResNet-18 (63.57% overall, 51.20% minority) slightly outperforms ConvNet (61.44%, 50.87%), while on `CIFAR-100` ConvNet (27.23%, 8.18%) marginally surpasses ResNet-18 (25.93%, 7.58%). The choice of backbone causes only modest differences in relative performance, confirming that our conclusions are not sensitive to architecture selection. The use of different standard backbones (ResNet-18 following Wang et al. (2022a) for `CIFAR-10`, and ConvNet following Wang et al. (2022a) for `CIFAR-100`) ensures consistency with prior work while providing sufficient model capacity at reasonable training cost.

**Real-World Dataset on PASCAL VOC.** To complement the experiments on synthetically corrupted CIFAR benchmarks, we evaluate CLAPOR on the PASCAL VOC dataset, a widely used real-world image classification benchmark. Following the IPLL protocol of Hong et al. (2023), we use the imbalanced VOC-PLL setting and further introduce annotation unreliability by randomly removing correct labels from 30% of instances (VOC-PLL with 30% noise). Table 15 reports the per-class size accuracy (many/medium/few), which provides finer-grained insight than overall accuracy alone. On VOC-PLL, CLAPOR achieves 42.25% overall accuracy and 19.42% on few-shot classes, significantly outperforming SOLAR (35.62% / 5.58%), PRODEN (10.15% / 0.00%), and PICO+ (7.70% / 0.00%). Under 30% noise, CLAPOR continues to lead with 20.20% overall and 8.92% on few-shot classes, while other baselines collapse severely. These results confirm that CLAPOR generalizes beyond synthetic I-UPLL settings to practical scenarios with real-world class imbalances and annotation noise.

**Training Time Analysis.** We report the wall-clock training time per epoch for CLAPOR and three representative baselines on `CIFAR-10` ($\rho = 50, \gamma = 0.3, \iota = 0.3$), evaluated with batch size 256 on a single NVIDIA GeForce RTX 4090 GPU. Table 16 shows the per-epoch training time alongside the corresponding overall and minority-class test accuracy. Handling the more challenging I-UPLL setting naturally requires additional computation: CLAPOR consumes approximately 45.00 seconds per epoch compared to 15.51 s for SOLAR, 7.94 s for PRODEN, and 9.66 s for PICO+. This overhead stems from the bi-level optimization that requires maintaining virtual parameters and performing an additional gradient update.

*Table 11.* Ablation study on the Dirichlet concentration parameter construction strategy on `CIFAR-10` ($\rho = 50, \gamma = 0.3, \iota = 0.3$).

| Method | Construction | Overall Acc. | Minority Acc. |
|---|---|---|---|
| CLAPOR-Uniform | $\boldsymbol{\alpha} = \mathbf{1}$ | 63.06 | 44.00 |
| CLAPOR-Random | $\alpha_j = \lambda \cdot u_j + \beta \ (u_j \sim \mathrm{Uniform}(0,1))$ | 60.75 | 48.77 |
| CLAPOR-Direct | $\boldsymbol{\alpha} = \lambda \cdot \hat{\boldsymbol{p}} + \beta$ | 62.44 | 54.13 |
| CLAPOR | $\boldsymbol{\alpha} = \lambda \cdot \mathit{Reverse}(\hat{\boldsymbol{p}}) + \beta$ | **63.48** | **62.26** |

*Table 12.* Extended sensitivity analysis of the concentration parameter $\beta$ on `CIFAR-10` ($\rho = 50, \gamma = 0.3, \iota = 0.3$). Best values among the displayed settings are in bold.

| $\beta$ | 0.25 | 0.5 | 1 | 2 | 3 | 4 | 5 | 10 | 20 | 30 | 40 | 50 |
|---|---|---|---|---|---|---|---|---|---|---|---|---|
| Overall Acc. | 56.93 | 59.30 | 56.10 | 58.61 | 58.66 | 57.05 | 58.68 | 59.76 | **62.28** | 59.63 | 58.44 | 59.50 |
| Minority Acc. | 47.23 | 45.60 | 41.93 | 44.80 | 45.70 | 45.63 | 45.23 | 48.50 | **58.10** | 50.43 | 42.03 | 45.03 |

Importantly, CLAPOR converges within approximately 200 epochs, while the 1000-epoch upper bound was chosen solely to ensure that all baselines have sufficient training budget for fair comparison. Overall, the moderate time cost is justified by the substantial performance gains, including a 22.80-percentage-point improvement in minority-class accuracy over the second-best method.

*Table 13.* Performance comparison on CIFAR-10 under reliable labels ($\gamma = 0, \rho = 100, \iota = 0.3$).

| Method | Overall Acc. | Minority Acc. |
|---|---|---|
| CLAPOR | **82.21** | **75.67** |
| PRODEN | 74.95 | 63.90 |
| CC | 74.22 | 67.27 |
| PICO | 73.92 | 64.30 |
| LWS | 42.97 | 0.00 |

*Table 14.* Backbone comparison for CLAPOR on CIFAR-10 and CIFAR-100.

| Dataset | Backbone | Overall Acc. | Minority Acc. |
|---|---|---|---|
| CIFAR-10 | ConvNet | 61.44 | 50.87 |
| | ResNet-18 | **63.57** | **51.20** |
| CIFAR-100 | ConvNet | **27.23** | **8.18** |
| | ResNet-18 | 25.93 | 7.58 |

*Table 15.* Performance of compared approaches on PASCAL VOC. "Many / Medium / Few" denotes per-class size accuracy.

| Setting | Method | Overall Acc. | Many Acc. | Medium Acc. | Few Acc. |
|---|---|---|---|---|---|
| VOC-PLL | CLAPOR | **42.25** | **67.25** | **40.62** | **19.42** |
| | SOLAR | 35.62 | 61.42 | 38.81 | 5.58 |
| | PRODEN | 10.15 | 33.25 | 0.44 | 0.00 |
| | PICO+ | 7.70 | 25.67 | 0.00 | 0.00 |
| VOC-PLL +30% noise | CLAPOR | **20.20** | **35.50** | **17.19** | **8.92** |
| | SOLAR | 9.65 | 32.08 | 0.06 | 0.00 |
| | PRODEN | 7.15 | 23.83 | 0.00 | 0.00 |
| | PICO+ | 6.47 | 21.58 | 0.00 | 0.00 |

*Table 16.* Training time per epoch and corresponding performance for CLAPOR and baseline methods. Batch size: 256. GPU: NVIDIA RTX 4090.

| Method | Time / Epoch (s) | Overall Acc. | Minority Acc. |
|---|---|---|---|
| CLAPOR | 45.00 | **58.00** | **49.07** |
| SOLAR | 15.51 | 56.59 | 26.27 |
| PRODEN | 7.94 | 35.11 | 20.50 |
| PICO+ | 9.66 | 35.16 | 13.13 |

