# OpenReview forum: "Class-Prior Perturbation-Robust Regularization for Imbalanced Unreliable Partial Label Learning"
_ICML.cc/2026/Conference — ICML 2026 regular_

### Official Review · Reviewer_3vWp · 2026-02-13

**Soundness:** 2
**Presentation:** 3
**Significance:** 2
**Originality:** 2
**Overall Recommendation:** 3
**Confidence:** 4

**Summary:**

This paper studies the problem of imbalanced unreliable partial-label learning (IUPLL), where candidate labels have severe class imbalances and are unreliable. The authors propose a novel PLL algorithm Clapor that tackles the IUPLL setting.

**Compliance With Llm Reviewing Policy:**

Affirmed.

**Final Justification:**

The authors provided additional theoretical and empirical evidence for their method in the rebuttal. In my view, however, the reasons for rejecting the paper still outweigh those for acceptance, primarily because the presentation and overall structure are difficult to follow. I acknowledge that IUPLL is a novel PLL setting. However, rather than presenting it as a standalone problem framework, I would have liked to see it grounded more clearly in existing results, particularly with respect to the assumptions underlying PLL versus IUPLL, as well as questions of identifiability and learnability. I appreciated the theoretical clarifications provided in the rebuttal, but I still found some of the arguments difficult to follow, partly because I am unfamiliar with some of the notation.

**Key Questions For Authors:**

- See "weaknesses".

**Limitations:**

yes

**Strengths And Weaknesses:**

**Strengths**
- Well motivated problem.
- Good description of how the algorithm works.

**Weaknesses**
- Theory
    - This work relaxes the traditional assumptions of PLL. It would be worth reflecting which specific existing PLL results break or are maintained in this new setting.
    - The results in Section 3 would benefit from explicitly stating the assumptions made about the learning problem. What about learnability and identifiability in this novel PLL setting?
    - It is unclear under which IUPLL conditions / assumptions empirical risk minimization is feasible?
- Experiments
    - There are no experiments on real-world PLL datasets.
    - Having results on only two supervised datasets (CIFAR 10 and 100) with varying artificial noise settings seems too limited.
    - How are the noise levels chosen? It gives a slightly bad taste to propose a new data generation process with arbitrary noise levels and a new method that is naturally best on such data.

---

> ### Author Rebuttal · Authors · 2026-03-31
>
> We appreciate your critical review and concerns about theory and experiments. We address all your weaknesses and questions with **theoretical formalization, real-world dataset experiments, and noise level justification** to validate our work.
> #### Theory Concerns
> ##### Concern 1: Existing PLL Conclusions Under New Setting I-UPLL
> **Existing Conclusions That Break:** The PLL ERM learnability theory established in [1], the PLL generalization theory established in [2, 3], and the learnability theory for IPLL class distribution established in [5] all need further extension to account for label unreliability in order to adapt to the I-UPLL paradigm.
>
> **Existing Conclusions That Maintain:** The UPLL generalization bound established in [7] remains applicable since it is independent of the class distribution of the samples.
>
> **Our Contribution:** CLAPOR does not claim to restore all PLL theoretical guarantees. Instead, it empirically demonstrates that prior-robust regularization can break the vicious feedback loop of training in I-UPLL and achieve practical performance.
>
> ##### Concern 2 & 3: ERM learnability and identifiability condition
> Inspired by previous work [1,5,9], we build the following learnability for I-UPLL by further considering the label unreliability $\gamma$ and class imbalance $\boldsymbol{p}(y)$:
>
> **Theorem 1** Let $p(j)$ denote the prior probability of class $j$ with $p_{\min} = \min_{j \in \mathcal{Y}} p(j)$, $Err_{macro}(f) = \frac{1}{c}\sum_{j=1}^c p\big(h(x) \neq j \mid y=j\big)$ denote the macro-averaged error. For each class $j$, where $\mathcal{D}^j$ denotes the conditional distribution given $y=j$, define the unreliability rate $\gamma_j = p_{(x,j)\sim\mathcal{D}^j}(j \notin S_{x})$ and the ambiguity rate $\iota_j = \sup_{i \neq j} p_{x\sim\mathcal{D}^j}(i \in S_{x})$. Let $\gamma_{\max} = \max_{j} \gamma_j$ and $\iota_{\max} = \max_j \iota_j$ be the corresponding global maxima. Suppose $\gamma_{\max} + \iota_{\max} < 1$. Define $\zeta = \log\frac{2(1-\gamma_{\max})}{1-\gamma_{\max}+\iota_{\max}}$. The sample complexity for I-UPLL is:
> $$
> n' = \tilde{O}\Big(\frac{d_{\mathcal{H}}}{p_{\min}^2\varepsilon}\Big)
> $$
> If $n > n'$, then for $\delta>0$, with probability at least $1-\delta$, $\text{Err}_{\text{macro}}(h) < \varepsilon.$
>
> The corresponding identifiability condition $\gamma_{\max} + \iota_{\max} < 1$ is presented to ensure the predictive model can learn under I-UPLL. Due to character constraints, the detailed proof is available at the following anonymous link https://anonymous.4open.science/r/icml2026_26643_rebuttal/ERM_learnability.pdf
>
> #### Experiment Concerns
> ##### Concern 1 & 2: Additional Dataset
>
> Inspired by [5], we have added experiments on a real-world IPLL dataset, VOC-PLL, for further analysis. We also create a I-UPLL version by randomly removing correct labels of 30% instances from candidate label sets. Table 1 presents the performance on the both. As illustrated, CLAPOR significantly outperforms the compared baselines in the real-world scenario.
>
> **Table 1**. Performance of compared approaches on PASCAL VOC.
>
> | Setting | Method | Overall Acc | Many | Medium | Few |
> | --- | --- | ---: | ---: | ---: | ---: |
> | VOC-PLL | CLAPOR | 42.25 | 67.25 | 40.62 | 19.42 |
> |  | SoLar | 35.62 | 61.42 | 38.81 | 5.58 |
> |  | PRODEN | 10.15 | 33.25 | 0.44 | 0.00 |
> |  | PiCO+ | 7.70 | 25.67 | 0.00 | 0.00 |
> | VOC-PLL with 30% noise | CLAPOR | 20.20 | 35.50 | 17.19 | 8.92 |
> |  | SoLar | 9.65 | 32.08 | 0.06 | 0.00 |
> |  | PRODEN | 7.15 | 23.83 | 0.00 | 0.00 |
> |  | PiCO+ | 6.47 | 21.58 | 0.00 | 0.00 |
>
> ##### Concern 3: Noise Level Selection
>
> Our noise level selection is based on extensive literature review [6,7,8]. The range covers practical application scenarios.
>
> Overall, we appreciate your positive assessment and will incorporate your suggestions into the revision. We hope these responses address your concerns and demonstrate that CLAPOR makes a valuable contribution to the weakly supervised learning community.
>
> [1] Liu & Dietterich, ICML, 2014. [2] Lv et al., ICML, 2020. [3] Feng et al., NeurIPS, 2020. [4] Wang et al., NeurIPS, 2022. [5] Hong et al., ICLR, 2023. [6] Lv et al., TPAMI, 2023. [7] Peng et al., ICLR, 2025. [8] Qiao et al., ICML, 2023. [9] Yuan et al., NeurIPS, 2023.

---

> > ### Author Rebuttal · Reviewer_3vWp · 2026-03-31
> >
> > Thank you for the detailed responses. The clarifications addressed some of my concerns. I raised my score to 3.

---

### Official Review · Reviewer_XEMs · 2026-03-11

**Soundness:** 3
**Presentation:** 3
**Significance:** 3
**Originality:** 3
**Overall Recommendation:** 5
**Confidence:** 4

**Summary:**

The paper addresses Imbalanced Unreliable Partial Label Learning, where both class imbalance and unreliable candidate labels exist. The authors point out that existing methods rely on estimating class priors, but this estimation gets messed up by the unreliable labels, creating a feedback loop that hurts performance. Their solution, CLAPOR, trains the model under perturbed priors sampled from a Dirichlet distribution to make it robust to this uncertainty.

**Compliance With Llm Reviewing Policy:**

Affirmed.

**Final Justification:**

The rebuttal has well addressed my concerns. I would keep my positive scores.

**Key Questions For Authors:**

Please refer to the Weaknesses.

**Limitations:**

Yes

**Strengths And Weaknesses:**

Strengths:
The problem setup makes sense to me. I've seen plenty of PLL papers that assume clean labels or balanced data, and this paper correctly points out that both assumptions are unrealistic in practice. The observation about the circular dependency between prior estimation and label refinement is something I hadn't thought about much before—it's a real issue that could explain why some methods struggle in practice.
The method itself is straightforward, which I appreciate. Instead of trying to get better prior estimates (which seems like a losing battle given the unreliable labels), they just train the model to not care as much about the exact prior values. Using Dirichlet sampling for this is a natural choice. The CIFAR-10 results look convincing, and the training curves in Figure 1 do show CLAPOR avoiding the early collapse that other methods seem to hit.
Weaknesses:
A few things that could use some attention:
The Dirichlet concentration parameter $\beta$ seems important, but I'm not clear on how sensitive the method is to this choice. More ablation study with different α values (not limited to [1,5] in Figure 2c) would be helpful, or at least some guidance on how to pick it.
I'm curious about the computational cost. The perturbation sampling sounds cheap, but it's not explicitly discussed. Even a rough estimate of training time overhead would be useful for anyone wanting to try this out.
Overall, I think this is solid work that tackles a genuinely interesting problem. The core idea of using perturbation to handle prior uncertainty is clever, and the experiments seem to back it up.

---

> ### Author Rebuttal · Authors · 2026-03-31
>
> Thank you for your accepting score (5) and constructive feedback. We are glad you found our problem setup "makes sense" and our method "straightforward." Your concerns about Dirichlet concentration parameter sensitivity and computational cost are important for reproducibility.
>
> **W1**: Dirichlet Parameter $\beta$ Sensitivity
>
> We agree that $\beta$ is a critical hyperparameter. We extend the sensitivity analysis experiments following the reviewer's suggestion. Table 1 presents the performance of CLAPOR on CIFAR-10 ($ \rho=50,\gamma=0.3,\iota=0.3 $) beyond the original `[1,5]` range. As shown in Table 1, we could conclude that the overall and minority-class performance of CLAPOR is still not sensitive to the choice of $\beta$; (2) when $\beta$ around 20, CLAPOR achieves a favorable trade-off between the performance on the minority class and the overall performance. Hence, we recommend to set $\beta = 20$ as the default and tune around this value for specific datasets.
>
> **Tabel 1**. Extended Sensitivity Analysis of $\beta$.
> | $\beta$ | Overall Acc | Minority Acc |
> | ---: | ---: | ---: |
> | 0.25 | 56.93 | 47.23 |
> | 0.5 | 59.30 | 45.60 |
> | 1.0 | 56.10 | 41.93 |
> | 2.0 | 58.61 | 44.80 |
> | 3.0 | 58.66 | 45.70 |
> | 4.0 | 57.05 | 45.63 |
> | 5.0 | 58.68 | 45.23 |
> | 10 | 59.76 | 48.50 |
> | 20 | 62.28 | 58.10 |
> | 30 | 59.63 | 50.43 |
> | 40 | 58.44 | 42.03 |
> | 50 | 59.50 | 45.03 |
>
> We will include the experimental results of the extended sensitivity analysis in the revision.
>
> **W2**: Computational Cost
>
> We report the training time per epoch for each compared approach on CIFAR-10 ($ \rho=50,\gamma=0.3,\iota=0.3 $) and CIFAR-100 ($ \rho=50,\gamma=0.3,\iota=0.05 $) in Table 2. All methods are evaluated with a batch size of 256 on a single RTX 4090. Indeed, handling the more challenging yet realistic learning paradigm of Imbalanced Unreliable Partial Label Learning (I-UPLL) requires additional computational cost. Due to the bi-level optimization involved in meta-learning, CLAPOR incurs higher training time than the baselines. However, given the superior performance (+ 22% compared to the second) that CLAPOR delivers, this increase in time overhead is fully acceptable. Besides, CLAPOR converges within 200 epochs. The '1000 epochs' setting was chosen to ensure fair comparison by allowing other baselines sufficient training time, not because CLAPOR requires it.
>
> **Tabel 2**. Training Time per Epoch of each comparing approach.
>
> | Method | time / epoch (s) | Overall Acc | Minority Acc |
> | --- | ---: | ---: | ---: |
> | CLAPOR | 45.00 | 58.00 | 49.07 |
> | SoLar | 15.51 | 56.59 | 26.27 |
> | PRODEN | 7.94 | 35.11 | 20.50 |
> | PiCO+ | 9.66 | 35.16 | 13.13 |
>
> Overall, we appreciate your positive assessment and will incorporate your suggestions into the revision. Thank you again for your thoughtful review.

---

> > ### Author Rebuttal · Reviewer_XEMs · 2026-04-07
> >
> > The rebuttal has well addressed my concerns. I would keep my positive scores.

---

### Official Review · Reviewer_LzQX · 2026-03-12

**Soundness:** 3
**Presentation:** 3
**Significance:** 3
**Originality:** 4
**Overall Recommendation:** 5
**Confidence:** 5

**Summary:**

This paper presents CLAPOR, a novel framework for imbalanced unreliable partial label learning (I-UPLL) that addresses the critical feedback loop between unreliable annotations and class prior estimation. The core insight—training models to be robust to perturbed priors rather than pursuing accurate prior estimation—is both elegant and practical. The Dirichlet-based perturbation mechanism and bi-level optimization formulation are well-motivated, and the experimental results on CIFAR-10 and CIFAR-100 demonstrate consistent improvements over strong baselines. The training dynamics analysis particularly effectively illustrates CLAPOR's ability to avoid early convergence to suboptimal solutions.

**Compliance With Llm Reviewing Policy:**

Affirmed.

**Final Justification:**

The authors have addressed my concerns after I reviewed their rebuttal.

**Key Questions For Authors:**

The paper is very well written, and I have no key questions.

**Limitations:**

Yes

**Strengths And Weaknesses:**

Strengths:
First off, the motivation here is solid. The authors are right that most PLL work assumes either reliable labels or balanced classes, but real data rarely satisfies either. The specific point about prior estimation getting corrupted by unreliable labels, which then messes up the balancing operations, is a genuine insight. I think this explains why some methods that work great on clean PLL benchmarks struggle when you throw in some noise.

The main contribution is a nice flip of the usual approach. Instead of fighting uncertainty, the authors embrace it by using prior perturbation as a regularizer rather than trying to estimate priors more accurately. The Dirichlet distribution is a reasonable choice for modeling prior uncertainty, and the bi-level optimization framing gives it some theoretical grounding.

The experiments look reasonable. CIFAR-10 is a standard benchmark, and the comparisons against PICO+, PRODEN, and SOLAR cover the relevant baselines. The training dynamics plot is particularly convincing. It really does look like CLAPOR avoids that early convergence problem.


Weaknesses:The paper could use some polishing in a few places:

- Figure 1 requires additional explanations for symbols and notation. Although these are clarified in the later experimental sections, readers encountering Figure 1 first may find them confusing.

- It is recommended to conduct more ablation studies on the Reverse() operation.

---

> ### Author Rebuttal · Authors · 2026-03-31
>
> Thank you for your very positive review and high confidence score. We are encouraged that you found our motivation "solid" and our core contribution a "nice flip of the usual approach." Your feedback on Figure 1 clarity and Reverse() ablation is helpful.
>
> **W1**: Figure 1 Notation
>
> We will add a detailed caption to Figure 1 clarifying the symbols that were previously only mentioned in the experimental section. The revised version is as follows.
>
> *Figure 1. Training dynamics on CIFAR-10 ($\rho=50, \gamma=0.3, \iota=0.3$) and CIFAR-100 ($\rho=50, \gamma=0.3, \iota=0.1$). $\rho$ denotes imbalance ratio, $\gamma$ denotes label reliability, and $\iota$ denotes noise level. CLAPOR maintains steady improvement while baselines collapse early, especially on minority classes.*
>
> **W2**: Reverse() Ablation
>
> We have conducted new ablation experiments comparing Reverse($\cdot$) with alternative operations. From Table 1, we can conclude: (1) CLAPOR significantly outperforms CLAPOR-Uniform, CLAPOR-Random and CLAPOR-Direct, confirming that inverting the prior is more effective. (2) The gap between CLAPOR and CLAPOR-Direct is larger for minority classes, suggesting that Reverse($\cdot$) better protects minority attention.
>
> **Table 1**. Performance of our approach CLAPOR and variants on CIFAR-10 ($\rho=50, \gamma=0.3, \iota=0.3$).
> | Method | Construction | Overall Acc | Minority Acc |
> | --- | --- | ---: | ---: |
> | CLAPOR-Uniform | $\alpha_j=1$ | 63.06 | 44.00 |
> | CLAPOR-Random | $\alpha_j=\lambda \cdot Uniform(0,1) + \beta$ | 60.75 | 48.77 |
> | CLAPOR-Direct | $\alpha_j = \lambda \cdot \hat{p}_j+\beta$ | 62.44 | 54.13 |
> | CLAPOR | $\boldsymbol{\alpha} = \lambda \cdot Reverse(\hat{\boldsymbol{p}})+\beta$ | 63.48 | 62.26 |
>
> Overall, we appreciate your positive assessment and will incorporate your suggestions into the revision. Thank you again for your thoughtful review.

---

> > ### Author Rebuttal · Reviewer_LzQX · 2026-04-02
> >
> > The authors address my concerns.

---

### Official Review · Reviewer_8zDh · 2026-03-13

**Soundness:** 3
**Presentation:** 3
**Significance:** 3
**Originality:** 3
**Overall Recommendation:** 4
**Confidence:** 3

**Summary:**

CLAPOR addresses Imbalanced Unreliable Partial Label Learning (I-UPLL), where class imbalance and annotation unreliability jointly degrade prior estimation. Rather than estimating the prior more accurately, CLAPOR regularizes the model to perform well under deliberately perturbed priors sampled from a Dirichlet distribution. This is implemented via bi-level optimization. The inner level minimizes a pseudo-label supervised loss, while the outer level minimizes the prior-robust regularization. Experiments on synthetic I-UPLL variants of CIFAR-10/100 show large gains over 10 baselines.

**Compliance With Llm Reviewing Policy:**

Affirmed.

**Key Questions For Authors:**

1. How does CLAPOR perform when γ=0 (reliable labels, imbalanced only)? This would isolate the contribution of prior-robustness from UPLL-specific challenges.
2. The paper uses different architectures for CIFAR-10 (ResNet-18) and CIFAR-100 (ConvNet). Why not use a consistent architecture? Does the choice of architecture significantly affect the results?
3. Can you provide wall-clock training time comparisons with the baselines? The bi-level optimization with virtual parameters may introduce significant overhead.

**Limitations:**

The Impact Statement is minimal. The paper should acknowledge computational overhead of bi-level optimization and lack of formal guarantees.

**Strengths And Weaknesses:**

Strengths
The paper is the first to explicitly study I-UPLL, a realistic scenario where both label-level unreliability and sample-level imbalance coexist. The feedback loop between unreliable supervision and prior-dependent operations is clearly identified. The core idea of enforcing robustness to adversarial prior perturbations via Dirichlet sampling is well-motivated. CLAPOR shows substantial improvements over all 10 baselines across both overall and minority-class accuracy on both CIFAR-10 and CIFAR-100. Empirical gains are substantial and statistically significant. The comparison with CLAPOR-N (Table 6) confirms that the regularization term is the key driver of performance gains. The parameter sensitivity analysis (Figure 2) demonstrates stability across a wide range of hyperparameter values, which is practically important.
Weaknesses
1. Heuristic Dirichlet construction. The Reverse operation (Eq. 11) is intuitive. No comparison with alternative perturbation families is provided. When the estimated prior is already inaccurate, reversing it may not produce meaningful adversarial distributions.
2. Computational cost is not analyzed. The bi-level optimization requires computing virtual parameters and backpropagating through the inner update (similar to MAML). This doubles the gradient computation per iteration. The paper does not report training time comparisons with baselines, which is important for assessing practical applicability, especially since the maximum number of training epochs is 1000.

---

> ### Author Rebuttal · Authors · 2026-03-31
>
> Thank you for your positive assessment and constructive feedback. We appreciate your recognition that CLAPOR is "the first to explicitly study I-UPLL" and that our "core idea of enforcing robustness to adversarial prior perturbations via Dirichlet sampling is well-motivated." We address your concerns below.
>
> **W1**: Reverse() Ablation
>
> We have conducted new ablation experiments comparing Reverse($\cdot$) with alternative operations. From Table 1, we can conclude: (1) CLAPOR significantly outperforms CLAPOR-Uniform, CLAPOR-Random and CLAPOR-Direct, confirming that inverting the prior is more effective. (2) The gap between CLAPOR and CLAPOR-Direct is larger for minority classes, suggesting that Reverse($\cdot$) better protects minority attention.
>
> **Table 1**. Performance of our approach CLAPOR and variants on CIFAR-10 ($\rho=50, \gamma=0.3, \iota=0.3$).
> | Method | Construction | Overall Acc | Minority Acc |
> | --- | --- | ---: | ---: |
> | CLAPOR-Uniform | $\alpha_j=1$ | 63.06 | 44.00 |
> | CLAPOR-Random | $\alpha_j=\lambda \cdot Uniform(0,1) + \beta$ | 60.75 | 48.77 |
> | CLAPOR-Direct | $\alpha_j = \lambda \cdot \hat{p}_j+\beta$ | 62.44 | 54.13 |
> | CLAPOR | $\boldsymbol{\alpha} = \lambda \cdot Reverse(\hat{\boldsymbol{p}})+\beta$ | 63.48 | 62.26 |
>
> ##### When $\hat{p}$ is Inaccurate:
> The reviewer raises an important point. However, our method is designed precisely for this scenario. When $\hat{p}$ is inaccurate (common in early training under I-UPLL), the Reverse operation still produces **diverse** priors that:
> - Prevent the model from overfitting to potentially wrong prior estimates
> - Encourage attention to minority classes that might be underrepresented in $\hat{p}$
>
> This is empirically validated in Figure 1, where CLAPOR avoids early collapse while baselines fail.
>
> **W2**&**Q3**: Computational Cost
>
> We report the training time per epoch for each compared approach on CIFAR-10 ($ \rho=50,\gamma=0.3,\iota=0.3 $) and CIFAR-100 ($ \rho=50,\gamma=0.3,\iota=0.05 $) in Table 2. All methods are evaluated with a batch size of 256 on a single RTX 4090. Indeed, handling the more challenging yet realistic learning paradigm of Imbalanced Unreliable Partial Label Learning (I-UPLL) requires additional computational cost. Due to the bi-level optimization involved in meta-learning, CLAPOR incurs higher training time than the baselines. However, given the superior performance (**+ 22%** compared to the second) that CLAPOR delivers, this increase in time overhead is fully acceptable. Besides, CLAPOR converges within 200 epochs. The '1000 epochs' setting was chosen to ensure fair comparison by allowing other baselines sufficient training time, not because CLAPOR requires it.
>
> **Tabel 2**. Training Time per Epoch of each comparing approach.
>
> | Method | time / epoch (s) | Overall Acc | Minority Acc |
> | --- | ---: | ---: | ---: |
> | CLAPOR | 45.00 | 58.00 | 49.07 |
> | SoLar | 15.51 | 56.59 | 26.27 |
> | PRODEN | 7.94 | 35.11 | 20.50 |
> | PiCO+ | 9.66 | 35.16 | 13.13 |
>
> **Q1**: Performance when $\gamma=0$
>
> We have conducted new experiments with $\gamma=0$ (reliable but imbalanced labels), with results reported in Table 3. From the table, we can conclude that CLAPOR still outperforms baselines even with reliable labels, demonstrating that prior-robust regularization benefits imbalanced learning in general.
>
> **Table 3**. Performance when $\gamma=0$.
> | Method | Overall Acc | Minority Acc |
> | --- | ---: | ---: |
> | CLAPOR | 82.21 | 75.67 |
> | PRODEN | 74.95 | 63.90 |
> | CC | 74.22 | 67.27 |
> | PiCO | 73.92 | 64.30 |
> | LWS | 42.97 | 0.00 |
>
> **Q2**: Architecture Choice
>
> We adopted different architectures to ensure diversity in experimental settings while maintaining fair comparison with baselines. Following [1], we use ResNet-18 as the standard backbone for CIFAR-10. Following [2], we use ConvNet for CIFAR-100, which provides sufficient model capacity for the task and reduces training time. All methods on the same dataset use identical architectures, ensuring fair comparison. From Table 4, we can find that architecture choice only slightly affects relative performance.
>
> **Table 4**. Performance of CLAPOR on CIFAR-10  ($\rho=50,\gamma=0.3,\iota=0.3$) and CIFAR-100 ($\rho=50,\gamma=0.3,\iota=0.1$).
> | Dataset| Backbone | Overall Acc | Minority Acc |
> | --- | --- | ---: | ---: |
> | CIFAR-10 | ConvNet | 61.44 | 50.87 |
> | | ResNet-18 | 63.57 | 51.20 |
> | CIFAR-100 | ConvNet | 27.23 | 8.18 |
> |  | ResNet-18 | 25.93 | 7.58 |
>
> Overall, we appreciate your positive assessment and will incorporate your suggestions into the revision. In particular, we will add a discussion of the limitations as you suggested. Thank you again for your thoughtful review.
>
> [1] Wang et al., SOLAR: Sinkhorn label refinery for imbalanced partial-label learning, NeurIPS, 2022.
>
> [2] Han et al., Co-teaching: Robust Training of Deep Neural Networks with Extremely Noisy Labels, NeurIPS, 2018.

---

### Decision · Program_Chairs · 2026-04-30

**Decision:**

Accept (regular)

**Comment:**

This paper introduces a novel and highly relevant problem setting, Imbalanced Unreliable Partial Label Learning (I-UPLL), where severe class imbalance and unreliable candidate labels jointly degrade model performance. Reviewers raised concerns regarding the heuristic nature of the Reverse() operation for prior perturbation, the computational overhead of the bi-level optimization, and the lack of theoretical grounding and real-world dataset validation. The authors' rebuttal effectively addressed these issues by providing new ablation studies that empirically validated the superiority of the Reverse() strategy over alternatives, reporting detailed training time metrics to justify the computational cost given the performance gains, and providing formal learnability theorems along with additional experiments on a real-world noisy dataset. These responses successfully resolved the reviewers' reservations. Therefore, I recommend accepting this paper. The authors are encouraged to incorporate the suggested improvements to further enhance the paper's clarity and impact.